# CareBench: A Comprehensive Benchmark for Accuracy, Robustness, and Fairness in Multimodal Fusion of EHR and Chest X-Rays

## Abstract

Machine learning holds great promise for advancing clinical decision support, yet multimodal models remain difficult to translate due to missing modalities and fairness concerns. We present CareBench, a comprehensive benchmark for evaluating accuracy, robustness, and fairness in multimodal fusion of Electronic Health Records (EHR) and chest X-rays (CXR), built on standardized cohorts from MIMIC-IV and MIMIC-CXR. CareBench provides an open-source data pipeline, a unified modeling framework spanning unimodal and multimodal methods, and a rigorous evaluation protocol that extends beyond predictive accuracy. Our analyses reveal several important findings: multimodal fusion improves accuracy when modalities are complete, but benefits shrink under realistic missingness unless architectures are explicitly designed to handle partial inputs; performance varies across tasks, metrics, and architectures, with robustness emerging as a design-dependent property; and multimodality can exacerbate fairness disparities, particularly across admission types and age groups. By establishing the first benchmark that jointly evaluates accuracy, robustness, and fairness for clinical multimodal learning, CareBench lays the foundation for developing methods that are not only accurate but also reliable and equitable in real-world healthcare settings.

## 1 Introduction

Machine learning is increasingly transforming clinical decision-making, with models capable of forecasting disease onset (Venugopalan et al., 2021; El-Sappagh et al., 2020), stratifying patient risk (Boehm et al., 2022), and personalizing treatment pathways (Esteva et al., 2022). A key frontier in this domain is multimodal learning (Elsharief et al., 2025), which aims to create a holistic patient view by integrating heterogeneous data sources such as Electronic Health Records (EHR) and Chest X-rays (CXR). The fusion of rich longitudinal EHR data with critical diagnostic CXR imaging has shown great potential for improved accuracy in many downstream prediction tasks (Hayat et al., 2022; Yao et al., 2024).

However, the transition from algorithmic potential to real-world clinical utility is fraught with challenges stemming from the imperfect nature of clinical data (Zhang et al., 2022). Two fundamental barriers, in particular, hinder the responsible deployment of multimodal models. The first is **missing modality**. In routine clinical practice, not all data types are collected for every patient; for example, a CXR may not be ordered for every ICU admission. This is not an edge case but a prevalent condition: in our cohort derived from the MIMIC databases, we find that nearly 75% of ICU stays lack a relevant CXR. The second critical challenge is **algorithmic fairness**. Clinical datasets often contain historical biases, and models trained on them can learn to exhibit performance disparities across demographic subgroups defined by attributes such as sex or race. For any multimodal learning methods to be trusted and adopted, they must be not only accurate but also fair, ensuring they do not perpetuate or amplify existing health disparities.

Recently, benchmarking efforts have made valuable contributions in understanding the state of the art, such as YAIB (Water et al., 2024) for reproducible unimodal EHR analysis and MedMod (Elsharief et al., 2025) for multimodal tasks on paired clinical data. However, there are critical gaps that remain to be filled. First, existing multimodal benchmarks primarily focus on complete-case scenarios, largely overlooking the pervasive issue of modality absence. Second, while fairness has been

Table 1: Comparison of CareBench and existing benchmarks.

| Benchmark | Modalities | Multimodal? | # of Models | Accuracy? | Robustness? | Fairness? |
|---|---|---|---|---|---|---|
| Purushotham et al. (2018) | EHR | ✗ | 18 | ✔ | ✗ | ✗ |
| Harutyunyan et al. (2019) | EHR | ✗ | 7 | ✔ | ✗ | ✗ |
| Barbieri et al. (2020) | EHR | ✗ | 13 | ✔ | ✗ | ✗ |
| MIMIC-Extract (Wang et al., 2020) | EHR | ✗ | 5 | ✔ | ✗ | ✗ |
| Sheikhalishahi et al. (2020) | EHR | ✗ | 4 | ✔ | ✗ | ✗ |
| FIDDLE (Tang et al., 2020) | EHR | ✗ | 4 | ✔ | ✗ | ✗ |
| Clairvoyance (Jarrett et al., 2021) | EHR | ✗ | 7 | ✔ | ✔ | ✗ |
| RadFusion (Zhou et al., 2021) | EHR & CT | ✔ | 1 | ✔ | ✗ | ✔ |
| EHR-TS-PT (McDermott et al., 2021) | EHR | ✗ | 1 | ✔ | ✗ | ✗ |
| HiRID-ICU (Yèche et al., 2022) | EHR | ✗ | 6 | ✔ | ✔ | ✗ |
| EHRSHOT (Wornow et al., 2023) | EHR | ✗ | 2 | ✔ | ✗ | ✗ |
| PyHealth (Yang et al., 2023) | EHR & Waveforms & Text & CXR | ✔ | 25 | ✗ | ✗ | ✗ |
| MC-BEC (Chen et al., 2023) | EHR & Text & Waveforms | ✔ | 1 | ✔ | ✔ | ✔ |
| INSPECT (Huang et al., 2023) | EHR & CT & Text | ✔ | 1 | ✔ | ✗ | ✗ |
| MEDFAIR (Zong et al., 2023) | Imaging | ✗ | 11 | ✔ | ✗ | ✔ |
| YAIB (Water et al., 2024) | EHR | ✗ | 8 | ✔ | ✗ | ✗ |
| MedMod (Elsharief et al., 2025) | EHR & CXR | ✔ | 11 | ✔ | ✗ | ✗ |
| CareBench (ours) | EHR & CXR | ✔ | 15 | ✔ | ✔ | ✔ |

benchmarked for unimodal medical imaging (Zong et al., 2023), it remains critically underexplored in the multimodal EHR-CXR fusion context. The complex interplay between missing data and subgroup biases is not well understood, and the field lacks standardized benchmarks that allow for direct and fair comparison of methods under these realistic and challenging conditions.

To address these limitations, we present CareBench, a **c**omprehensive benchmark for **a**ccuracy, **r**obustness, and fair**ne**ss in multimodal fusion of EHR and chest X-rays. Our work provides three core components to the research community: (i) an open-source and reproducible **data extraction pipeline** for MIMIC-IV and MIMIC-CXR to establish a standard cohort for evaluation; (ii) a unified, open-source **modeling framework** implementing a wide array of models, from unimodal baselines to state-of-the-art fusion architectures, to facilitate fair comparison and future extensions; and (iii) a rigorous **evaluation protocol** that moves beyond standard predictive performance. Critically, our benchmark introduces extensive analyses of model **robustness** against varying ratios of missing modalities and **algorithmic fairness** across different patient subgroups. While fairness has been benchmarked for medical imaging in isolation (Zong et al., 2023), these crucial assessments for clinical translation have not been integrated into a multimodal EHR-CXR fusion context.

Through this benchmark, we revealed several key scientific insights, including: 1. Multimodal fusion improves accuracy with complete data but often fails under high missingness unless architectures are explicitly designed to handle incomplete inputs. 2. Model performance varies across tasks, metrics, and architectures. Robustness emerges as a design-dependent property. 3. Systematic fairness disparities exist across admission types and age groups, and multimodality can amplify these gaps, underscoring the need for subgroup-aware and fairness-aware approaches in clinical deployment.

## 2 RELATED WORK

The goal of this work is to establish a unified and extensible benchmark that spans unimodal and multimodal fusion algorithms while jointly evaluating accuracy, robustness, and fairness. Table 1 situates CareBench among prior benchmarks, showing that existing efforts are either unimodal or limited in multimodal scope. CareBench advances the field as the first EHR–CXR benchmark with a transparent data pipeline, diverse fusion models, and a tri-dimensional evaluation protocol.

**EHR Benchmarks in Healthcare** The widespread adoption of electronic health records (EHRs) has enabled the creation of large-scale datasets, spurring the development of numerous benchmarks for clinical prediction tasks. Purushotham et al. (2018) and Harutyunyan et al. (2019) introduced early EHR benchmarks on MIMIC-III, demonstrating the utility of deep learning models in clinical outcome prediction. Barbieri et al. (2020) extended this line by evaluating neural ODEs and attention-based models for readmission and patient risk stratification. Sheikhalishahi et al. (2020) compared machine learning models on the multi-center eICU dataset, highlighting generalization across healthcare systems. To address data accessibility, MIMIC-Extract (Wang et al., 2020) and FIDDLE (Tang et al., 2020) provided standardized preprocessing pipelines, while Clairvoyance (Jarrett et al., 2021) offered an end-to-end AutoML-friendly framework for medical time-series. More recent efforts

include EHR-TS-PT (McDermott et al., 2021) and EHRSHOT (Wornow et al., 2023), which explored pre-training and few-shot learning for EHR time series, as well as HiRID-ICU (Yèche et al., 2022), which benchmarked machine learning models on high-resolution ICU data. Finally, YAIB (Water et al., 2024) proposed a modular, multi-dataset EHR framework emphasizing extensibility. Despite their contributions, all these efforts focus exclusively on the EHR modality, whereas real-world clinical decision-making is inherently multimodal.

**Multimodal Benchmarks for Clinical Prediction**    Recognizing the benefits of integrating multi-modal data for clinical tasks, recent years have seen a growing number of multimodal benchmarks. INSPECT (Huang et al., 2023) and RadFusion (Zhou et al., 2021) established multimodal benchmarks for pulmonary embolism diagnosis and prognosis using CT and EHR data, though both adopted only late-fusion strategies. MC-BEC (Chen et al., 2023) introduced a multimodal benchmark for emergency care with EHR, notes, and waveforms, and uniquely assessed robustness to missing data and fairness. However, its fusion approach was limited to a simple late-fusion scheme, leaving advanced and adaptive fusion strategies unexplored. PyHealth (Yang et al., 2023) provided a comprehensive deep learning toolkit covering EHR, waveforms, text, and imaging; however, it does not provide standardized performance comparisons across models. Most relevant to our work, MedMod (Elsharief et al., 2025) introduced the first EHR–CXR benchmark, comparing early, joint, and late fusion paradigms. Yet, MedMod did not systematically evaluate robustness to missing modalities or fairness across subgroups. In parallel, MEDFAIR (Zong et al., 2023) focused on fairness benchmarking in imaging, but was limited to unimodal settings. Together, these works underscore the need for a more comprehensive benchmark.

**Multimodal Fusion for Clinical Prediction**    A growing body of work has explored multimodal fusion methods to address key challenges in clinical prediction, such as heterogeneous data distributions, irregular sampling, and missing modalities. Simple late fusion remains a widely used baseline, while more advanced approaches, including DAFT (Pölsterl et al., 2021), MMTM (Joze et al., 2020), and UTDE (Zhang et al., 2023), enable tighter cross-modal interactions under complete-modality settings. To cope with missing data, models such as HEALNet (Hemker et al., 2024), Flex-MoE (Yun et al., 2024), DrFuse (Yao et al., 2024), UMSE (Lee et al., 2023), and M3Care (Zhang et al., 2022) introduce mechanisms for flexible modality integration, disentangling shared and specific features, or imputing task-relevant latent representations. Collectively, these methods underscore the central importance of robust and adaptive fusion in clinical machine learning, motivating our systematic evaluation of state-of-the-art models within CareBench.

## 3 DATASET EXTRACTION

We constructed our benchmark using large-scale real-world ICU databases, specifically MIMIC-IV (Johnson et al., 2023) and MIMIC-CXR (Johnson et al., 2019). The former contains de-identified records of adult patients admitted to either intensive care units or the emergency department of Beth Israel Deaconess Medical Center (BIDMC) between 2008 and 2019, and the latter is a publicly available dataset of chest radiographs collected from BIDMC, where a subset of patients can be matched with those in MIMIC-IV.

### 3.1 COHORT CONSTRUCTION

We construct two cohorts of ICU stays from the MIMIC-IV database: a base cohort containing all ICU episodes that satisfy clinical and temporal consistency requirements, and a matched subset further restricted to encounters with paired chest radiographs. The detailed exclusion criteria used to construct the data cohorts can be found in Fig. 5.

**The Base Cohort**    Starting from the 73,181 ICU stays available from the MIMIC-IV database, we remove stays lacking essential clinical documentation (e.g., missing discharge notes or diagnostic codes) and episodes with implausible temporal records, such as hospital admission times occurring after ICU admission or discharge. To focus on clinically meaningful acute episodes, we further excluded ICU stays of less than 6 hours, and admissions labeled as non-urgent or elective, repeated ICU episodes within the same hospitalization. Since short ICU stays often represent observational or step-down care, and usually have insufficient longitudinal information for robust prediction, we further exclude ICU stays shorter than 48 hours to construct the base cohort, which eventually contains 26,947 ICU stays.

**The Matched Subset**    To establish a multimodal benchmark, we require the availability of at least one chest radiograph within a window spanning 24 hours before to 48 hours after ICU admission.

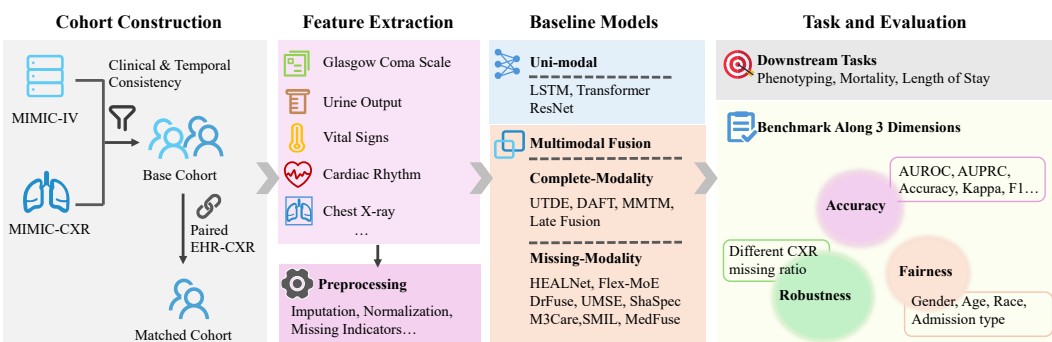

Figure 1: Overview of CareBench pipeline.

This yields a matched subset of 7,149 ICU stays, representing patients for whom both structured EHR data and chest radiographs are available.

## 3.2 FEATURE EXTRACTION

**EHR Feature Extraction** We extracted a comprehensive set of structured electronic health record (EHR) features from the MIMIC-IV v2.2 database. Our extraction pipeline was designed to capture clinically relevant variables across multiple physiological domains, including vital signs, neurological status (Glasgow Coma Scale), cardiac rhythm, respiratory support parameters ($O_2$ flow, $FiO_2$), fluid balance (urine output), and body weight. All features were retrieved using structured SQL queries executed on a locally deployed PostgreSQL instance of MIMIC-IV v2.2, with the Python toolkit sqlalchemy (Bayer, 2012) for database interaction. To ensure temporal alignment within each ICU stay, we joined relevant source tables, including chartevents, labevents, procedureevents, as well as derived modules such as gcs, kdigo_uo, ventilator_setting, blood_differential, weight_durations, and enzyme, using stay_id and timestamp synchronization. We initially explored 25 distinct categories of measurements. However, features with a missingness rate greater than 90% were empirically excluded. Furthermore, treatment-related variables, such as continuous renal replacement therapy (CRRT), invasive line placement, and mechanical ventilation settings, were removed to avoid potential label leakage. The final set of features used in CareBench is summarized in Table 4, with each variable annotated by category, source table, summary statistics, and missing rate.

**EHR Preprocessing** To maintain compatibility with most baseline models, and following prior MIMIC benchmarks (Harutyunyan et al., 2019; Elsharief et al., 2025), we resampled the EHR data at an hourly resolution. Missing values were imputed using forward filling and median imputation strategies, while binary cardiac rhythm indicators were directly imputed with 0. To preserve information on data availability, we additionally retained binary mask columns indicating the presence or absence of each measurement, as missingness itself can be informative in clinical settings (Morid et al., 2023). For continuous variables, we applied robust normalization using the median and interquartile range (IQR) to mitigate the influence of outliers.

**CXR Selection Criteria** To ensure temporal and clinical alignment between imaging and EHR data, we restricted the chest X-ray (CXR) cohort to scans acquired during the patient's current ICU stay. Only frontal-view images with an Anterior-Posterior (AP) projection were included, as this is the standard acquisition protocol for bedside radiography in critical care settings. Among all eligible AP views, we selected the most recent CXR prior to the prediction timepoint to best reflect the patient's latest cardiopulmonary status.

## 4 BENCHMARK DESIGN

### 4.1 MODELS

We benchmark a broad set of models for multimodal fusion of EHR and chest X-rays, spanning unimodal baselines, simple fusion strategies, and recent state-of-the-art multimodal algorithms that can be adapted to clinical settings. Detailed description of these baselines can be found in Section B.1.

**Uni-modal Baselines** We include uni-modal models as baselines to establish reference performance for each modality. For EHR, we include the classic Long Short-Term Memory network (LSTM) and the Transformer model. They are widely used architectures for capturing temporal dependencies in

sequential EHR data. For CXR, we adopt the ResNet-50 model, which is pretrained on ImageNet. These baselines quantify the stand-alone predictive value of each modality.

**Complete-Modality Multimodal Fusion Methods**   This group of models assumes that all modalities are present at both training and inference, including: UTDE (Zhang et al., 2023), DAFT (Pölsterl et al., 2021), MMTM (Joze et al., 2020), and Late Fusion.

**Missing-Modality Multimodal Fusion Methods**   We implement a broad set of multimodal fusion methods that could handle missing modalities, covering both models developed specifically for clinical data and models originally proposed in other domains (e.g., video–audio classification) that can be naturally adapted to clinical EHR–CXR fusion. This collection spans diverse design paradigms, including: HEALNet (Hemker et al., 2024), Flex-MoE (Yun et al., 2024), DrFuse (Yao et al., 2024), UMSE (Lee et al., 2023), ShaSpec (Wang et al., 2023), M3Care (Zhang et al., 2022), MedFuse (Hayat et al., 2022), and SMIL (Ma et al., 2021).

## 4.2 Downstream Tasks and Evaluations

We evaluate models on three downstream tasks that are highly relevant to clinical decision support: phenotyping classification, mortality prediction, and length-of-stay (LoS) prediction. All tasks use patient data observed within a fixed prediction window, and are evaluated on both the base cohort (realistic setting with missing modalities) and the matched subset (complete modalities). To ensure comparability, we adopt patient-level train/validation/test splits and report established metrics tailored to each task.

**Phenotyping Classification**   The goal of phenotyping is to predict the set of acute and chronic conditions present during an ICU stay. Following prior benchmarks, we construct 25 phenotypes derived from ICD-9 and ICD-10 diagnosis codes, spanning common comorbidities and critical conditions. The task is formulated as multi-label classification, requiring models to output a binary prediction for each phenotype simultaneously. To comprehensively evaluate model performance, we employ a suite of metrics including Area Under the Receiver Operating Characteristic Curve (AUROC), Area Under the Precision-Recall Curve (AUPRC), F1 score, precision, recall, specificity, and accuracy (ACC).

**Mortality Prediction**   This task focuses on predicting in-hospital mortality within the first 48 hours of ICU admission, formulated as a binary classification problem. The objective is to determine whether a patient will survive or die during hospitalization, enabling early identification of critically ill patients at high risk of deterioration. Labels are derived directly from hospital discharge status, with positive cases defined as patients who died during their hospital stay and negative cases as those who were discharged alive. To ensure clinically realistic evaluation, the prediction window is restricted to the initial 48 hours of an ICU stay, using only information available within that period. Model performance is assessed using a comprehensive set of metrics, including AUROC and AUPRC to capture threshold-independent discrimination ability, as well as F1-score, accuracy (ACC), precision, recall, and specificity.

**LoS Prediction**   Accurate estimation of ICU LoS is important for clinical planning and resource allocation. In this task, we use the first 48 hours of EHR and CXR data to predict the remaining hospital stay (RLOS). The RLOS is discretized into clinically meaningful intervals: 2–3 days, 3–4 days, 4–5 days, 5–6 days, 6–7 days, 7–14 days, and 14+ days, resulting in a multi-class classification problem with ordinal structure. Performance is evaluated using the ACC, F1 score, and Cohen's Kappa weighted quadratic, with additional metrics such as precision, recall, and specificity reported for completeness.

## 4.3 Implementation Details

We implement all models in Python 3.12.2 using PyTorch 2.5.1 and PyTorch Lightning 2.2, running with CUDA 12.1 and cuDNN 9.1.0. Experiments are conducted on servers equipped with AMD EPYC 7763 64-Core CPUs, 512 GB RAM, and 4×NVIDIA RTX 4090 GPUs (24 GB memory each). To ensure fair comparison, we adopt consistent training settings across tasks and perform Bayesian hyperparameter optimization to tune model-specific configurations.

**Hyperparameter tuning**   We employ Bayesian optimization with a Gaussian process surrogate (gp-minimize from scikit-optimize) and the gp-hedge acquisition strategy, which adaptively balances exploitation and exploration by combining multiple acquisition functions (LCB, EI, PI). Each search runs for 20 iterations, starting from 5 random initial configurations. For each candidate configuration, the framework launches full training runs on the specified search folds (here fold = 1), with three

Table 2: Predictive Performance over the matched subset.

| Models | Phenotyping | | | Mortality | | | Length of Stay | | |
|---|---|---|---|---|---|---|---|---|---|
| | AUPRC | AUROC | F1 | AUPRC | AUROC | F1 | ACC | Kappa | F1 |
| **LSTM** | 0.4520 | 0.7153 | 0.3279 | 0.4509 | 0.8032 | 0.3192 | 0.3973 | 0.1925 | 0.1864 |
| **Transformer** | 0.4718 | 0.7244 | 0.3395 | 0.4474 | 0.8208 | 0.2131 | 0.3921 | 0.1870 | 0.1923 |
| **ResNet** | 0.3997 | 0.6679 | 0.2693 | 0.2252 | 0.6879 | 0.0495 | 0.3463 | 0.1151 | 0.1333 |
| **UTDE** | 0.4920 | **0.7402** | 0.3029 | 0.4510 | 0.8228 | 0.2791 | 0.3945 | 0.1946 | 0.1930 |
| **DAFT** | 0.4799 | 0.7307 | 0.2750 | 0.4354 | 0.8342 | 0.0078 | 0.3937 | 0.1800 | 0.1604 |
| **MMTM** | 0.4723 | 0.7269 | **0.3808** | 0.3571 | 0.8007 | 0.3201 | 0.2607 | 0.0907 | 0.1648 |
| **LateFusion** | 0.4890 | 0.7381 | 0.2977 | 0.4328 | 0.8233 | 0.2871 | 0.3940 | 0.1910 | 0.1886 |
| **HEALNet** | 0.4714 | 0.7261 | 0.2104 | 0.4507 | 0.8356 | 0.1752 | 0.4036 | 0.1948 | 0.1866 |
| **Flex-MoE** | 0.4876 | 0.7355 | 0.2968 | 0.4734 | 0.8401 | 0.2664 | 0.3980 | 0.1946 | 0.1870 |
| **DrFuse** | **0.4928** | 0.7387 | 0.3762 | **0.4813** | 0.8378 | **0.3375** | **0.4060** | 0.1947 | 0.1847 |
| **UMSE** | 0.4462 | 0.7106 | 0.1916 | 0.3949 | 0.8014 | 0.0039 | 0.3912 | 0.1826 | 0.1602 |
| **ShaSpec** | 0.4813 | 0.7342 | 0.3414 | 0.4527 | 0.8331 | 0.2204 | 0.3980 | **0.1978** | **0.1936** |
| **M3Care** | 0.4881 | 0.7388 | 0.2750 | 0.4447 | 0.8277 | 0.2410 | 0.4003 | 0.1854 | 0.1602 |
| **MedFuse** | 0.4777 | 0.7317 | 0.2556 | 0.4717 | **0.8509** | 0.1569 | 0.4031 | 0.1942 | 0.1672 |
| **SMIL** | 0.4517 | 0.7130 | 0.3081 | 0.4532 | 0.8372 | 0.2641 | 0.3989 | 0.1791 | 0.1460 |

Table 3: Predictive Performance over the base cohort.

| Models | Phenotyping | | | Mortality | | | Length of Stay | | |
|---|---|---|---|---|---|---|---|---|---|
| | AUPRC | AUROC | F1 | AUPRC | AUROC | F1 | ACC | Kappa | F1 |
| **LSTM** | 0.4684 | 0.7547 | 0.3336 | 0.4797 | 0.8608 | 0.3668 | 0.4120 | 0.1966 | 0.1781 |
| **Transformer** | 0.4787 | 0.7591 | **0.3618** | 0.5042 | 0.8674 | **0.4024** | 0.4171 | 0.2040 | 0.1976 |
| **ResNet** | 0.2801 | 0.5650 | 0.1149 | 0.1339 | 0.5655 | 0.0013 | 0.3542 | 0.0408 | 0.1059 |
| **UTDE** | 0.4853 | 0.7636 | 0.2787 | 0.5029 | 0.8683 | 0.2864 | 0.4162 | 0.2071 | 0.2013 |
| **DAFT** | 0.4755 | 0.7565 | 0.2351 | 0.4952 | 0.8698 | 0.2297 | 0.4179 | 0.2080 | 0.1977 |
| **MMTM** | 0.4705 | 0.7553 | 0.3390 | 0.4749 | 0.8649 | 0.3720 | 0.4039 | 0.1714 | 0.1615 |
| **LateFusion** | 0.4847 | 0.7625 | 0.2891 | 0.4995 | 0.8672 | 0.3386 | 0.4161 | 0.2101 | 0.2001 |
| **HEALNet** | 0.4752 | 0.7578 | 0.2196 | 0.4914 | 0.8729 | 0.2911 | 0.4182 | 0.1998 | 0.1906 |
| **Flex-MoE** | 0.4835 | 0.7619 | 0.3162 | 0.5065 | 0.8684 | 0.3054 | 0.4137 | 0.2059 | 0.1879 |
| **DrFuse** | 0.4845 | **0.7639** | 0.3613 | 0.4999 | 0.8736 | 0.3561 | 0.4190 | 0.1965 | 0.1906 |
| **UMSE** | 0.4564 | 0.7482 | 0.1733 | 0.4337 | 0.8455 | 0.1507 | 0.4054 | 0.1842 | 0.1814 |
| **ShaSpec** | 0.4848 | 0.7626 | 0.3473 | 0.5000 | 0.8690 | 0.3882 | **0.4208** | 0.2088 | 0.1912 |
| **M3Care** | **0.4883** | 0.7637 | 0.2713 | 0.4994 | 0.8691 | 0.2928 | 0.4195 | 0.2044 | 0.1879 |
| **MedFuse** | 0.4686 | 0.7556 | 0.2204 | **0.5079** | **0.8741** | 0.3158 | 0.4177 | **0.2127** | **0.2035** |
| **SMIL** | 0.4474 | 0.7420 | 0.2964 | 0.4782 | 0.8596 | 0.3073 | 0.4171 | 0.1921 | 0.1570 |

random seeds (42, 123, 1234) to account for variance. Results from all seeds are aggregated by computing the mean and standard deviation of multiple metrics (ACC, AUPRC, AUROC, F1, etc.), and the task-specific selection criterion is applied: we maximize AUPRC for phenotyping and mortality, and ACC for LoS prediction.

**Hyperparameter search space**    To ensure fair comparison, we standardize the core training setup across all models (learning rate, batch size, epochs, early stopping, etc.) and restrict hyperparameter search to model-specific components. For several baselines, these fixed settings fully determine the configuration, leaving no tunable components. Consequently, hyperparameter search is only applied to models with explicit model-specific parameters, namely DrFuse, FlexMoE, HEALNet, M3Care, ShaSpec, and SMIL. An overview of their searched parameters and ranges is provided in Table 5, while full task-wise configurations and best hyperparameters obtained from Bayesian optimization are reported in the Appendix B.2.

## 5 RESULTS AND DISCUSSIONS

### 5.1 OVERALL PREDICTIVE PERFORMANCE

We first benchmark the predictive performance of all models on three distinct clinical tasks: phenotyping, mortality prediction, and length of stay (LoS) prediction. Our experiments, conducted on both an ideal-case matched subset with complete data and a realistic base cohort with significant modality absence, reveal several key findings regarding the efficacy and limitations of multimodal fusion. We analyze the comparative efficacy of different modeling paradigms under varying data availability, with full results presented in Table 2 and Table 3. See Section C for complete results.

**Insight 1: Multimodal fusion outperforms unimodal models on complete data.** On the matched subset, where all modalities are present for every patient (Table 2), the advantages of multimodal fusion are evident. For the phenotyping task, top-performing fusion models such as DrFuse (0.4928 AUPRC) and UTDE (0.7402 AUROC) demonstrate a significant improvement over the strongest unimodal baseline, the EHR Transformer (0.4718 AUPRC, 0.7402 AUROC). A similar trend holds for mortality prediction, where MedFuse achieves an AUROC of 0.8509, substantially outperforming the Transformer's 0.8208. For LoS prediction, multimodal models also show a modest benefit, with ShaSpec attaining the highest Kappa score (0.1978) compared to the best unimodal baseline (LSTM, 0.1925). These results confirm that when modalities are complete, CXR images provide complementary information to structured EHR data, which can be effectively leveraged to improve predictive accuracy. The success of diverse architectures indicates that their integration yields a richer patient representation that translates to superior clinical prediction.

**Insight 2: Specialized architectures are essential for leveraging incomplete multimodal data.** Evaluating models on the base cohort (Table 3), where nearly 73.5% of patients lack a CXR, highlights the critical and practical challenge of missing modality. The EHR-only Transformer establishes a strong benchmark that many multimodal models fail to surpass: the highest mortality F1-score (0.4024), and a LoS Kappa score (0.2040), outperforming several fusion methods designed for complete data, such as MMTM (0.3720 and 0.1714, respectively). This reveals a critical lesson: naively applying models designed for complete-case scenarios (e.g., DAFT, MMTM) does not guarantee a benefit and often fails once missingness is introduced. However, our results provide compelling evidence that architectures specifically designed to handle modality absence are essential to make full use of the multimodal data. The models explicitly tailored for this challenge, such as MedFuse, M3Care, and HEALNet, consistently outperform both the unimodal baseline and the complete-case fusion methods. For mortality prediction , MedFuse attains the highest AUROC (0.8741) and AUPRC (0.5079). In LoS prediction, MedFuse also leads with the best Kappa (0.2127) and F1-score (0.2035), finally surpassing the strong EHR Transformer. A similar advantage is seen in LoS prediction. This distinction is critical: only through purpose-built architectures can the theoretical benefits of fusion be reliably realized in the presence of missing data.

**Insight 3: There is no single best model across all tasks and metrics.** Our comprehensive benchmark shows that no single model is universally superior. The optimal choice depends on both the task and the evaluation metric. For example, in mortality prediction on the base cohort, MedFuse excels at AUROC and AUPRC, while the unimodal Transformer is the best for F1-score. In phenotyping, DrFuse offers the best AUROC, whereas M3Care leads on AUPRC, and for the LoS task, MedFuse tops Kappa and F1, while ShaSpec attains the highest accuracy. This heterogeneity highlights a key conclusion: the importance of moving beyond single-metric leaderboards. A robust evaluation framework must consider a suite of metrics that reflect diverse clinical priorities. Our findings advocate for a nuanced, context-aware approach to model selection and demonstrate the necessity of comprehensive benchmarking to accurately assess the state of the art.

### 5.2 ROBUSTNESS TO MODALITY MISSINGNESS

To further assess model robustness, we evaluate performance under varying degrees of a missing modality. To rule out the effect of sample size, we simulate this by starting with the complete, matched subset (0% missingness) and progressively increasing the ratio of missing CXRs up to 80%. The results are visualized in Fig. 2. For visual clarity, we only present the top six performing models for each task in the main text. The results reveal key insights into model behavior under data scarcity.

**Insight 4: The impact of missing modality is task-dependent.** The effect of a missing modality is highly contingent on the clinical task. For phenotyping classification, AUPRC consistently degrades as the CXR missingness ratio increases. This trend, coupled with widening standard deviations at

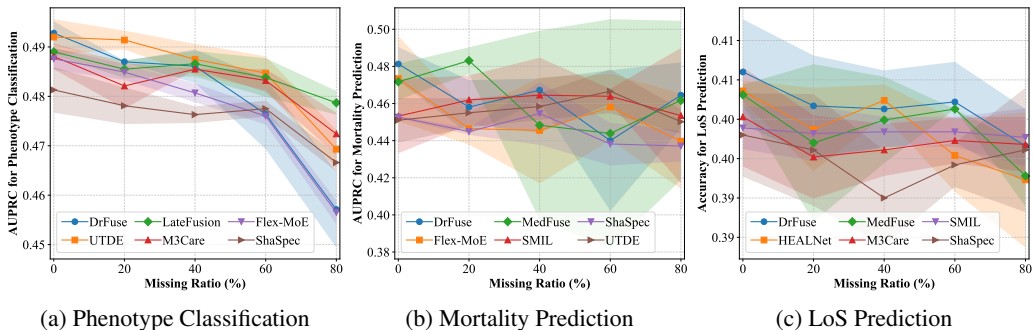

(a) Phenotype Classification      (b) Mortality Prediction      (c) LoS Prediction

Figure 2: Robustness of the top six performing methods for each task under varying ratios of modality missingness. The shaded areas represent standard deviations.

higher missingness levels for most models, indicates that visual features are highly informative for this task, and their absence not only compromises predictive power but also reduces model stability. In contrast, performance on mortality prediction and LoS prediction is more stable in terms of mean performance, exhibiting relatively flat trend lines. However, for mortality prediction, the variance bands widen noticeably with increased missingness, suggesting that model predictions become less consistent even as the average performance is maintained. This suggests that the mortality and LoS prediction tasks are predominantly reliant on the rich temporal data of the EHR.

**Insight 5: Model robustness is architecture-dependent.** The trajectories in Fig. 2 demonstrate that multimodal fusion architectures differ substantially in their resilience to different ratios of CXR-missingness. While performance degradation is inevitable as missing ratio increases, some models show smoother declines. For example, SMIL remains relatively stable in mortality and LoS prediction, likely due to its feature reconstruction and regularization mechanisms. In contrast, models such as DrFuse, ShaSpec, and Flex-MoE exhibit sharp drops under severe missingness (from 60% to 80%) in phenotype classification, reflecting the fragility of approaches that rely heavily on a shared latent representation as proxy for absent modalities. These findings suggest that robustness is not an inherent benefit of multimodality, but rather an outcome of architectural choices. Moreover, Fig. 2 reports only the top six performing methods, most of which already incorporate mechanisms for missingness, reinforcing that clinically deployable multimodal systems should explicitly support partial-modality inputs to enhance robustness under realistic patterns of missingness.

### 5.3 FAIRNESS

To assess fairness, we stratify model performance across four clinically relevant attributes: ICU admission type, gender, race, and age. We use AUPRC as the primary metric and quantify disparities with the AUPRC gap (difference between the best- and worst-performing subgroups), reporting results on both the matched subset and the base cohort. We include the detailed subgroup result on ICU admission type in Fig. 3 and performance gap distribution in Fig. 4 on phenotype classification. For mortality and LoS prediction, see Section C.3.

**Insight 6: Admission-type disparities reflect both clinical characteristics and subgroup prevalence.** Fig. 3 shows systematic performance gaps across admission types in phenotyping, with substantial implications once subgroup prevalence is considered. On the matched subset, Direct Emergency patients (3.65% of admissions) and Urgent patients (16.15%) achieve AUPRC above 0.50, while Observation Admits (10.95%) remain below 0.45. This gap reflects both the richer multimodal signals available for acute admissions and the sparse, lower-acuity data associated with observation cases. In the base cohort, where around 75% of patients lack CXR, Urgent and EW Emergency patients (together representing over 80% of admissions) maintain relatively strong AUPRC due to robust EHR signals, while Observation Admits remain the lowest-performing subgroup. These findings highlight a dual source of disparity: multimodal models are biased toward high-resource, high-prevalence admission types, while systematically underserving smaller but clinically meaningful groups such as Observation Admits and Direct Emergencies. Fairness evaluation should account for both subgroup gaps and prevalence, and multimodal models should adopt subgroup-aware training or reweighting to ensure equitable performance across admission types.

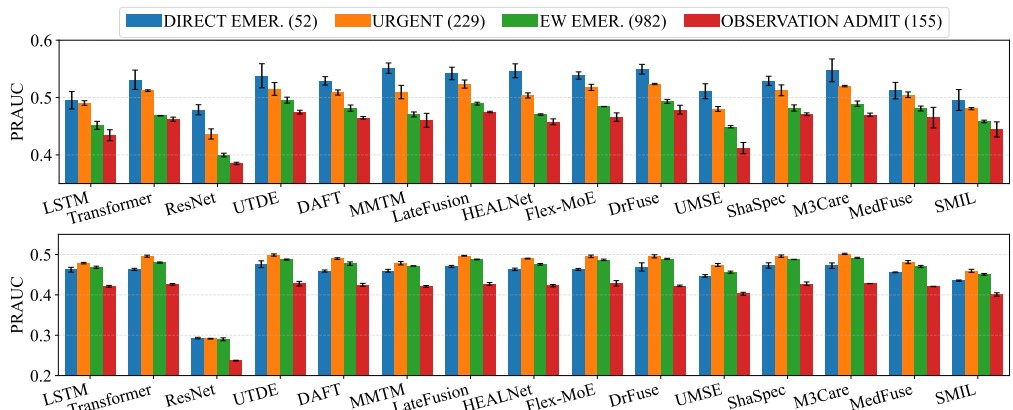

Figure 3: The AUPRC score for phenotype classification across different ICU admission types (with test set sub-group size) on the matched subset (upper row) and the base cohort (lower row).

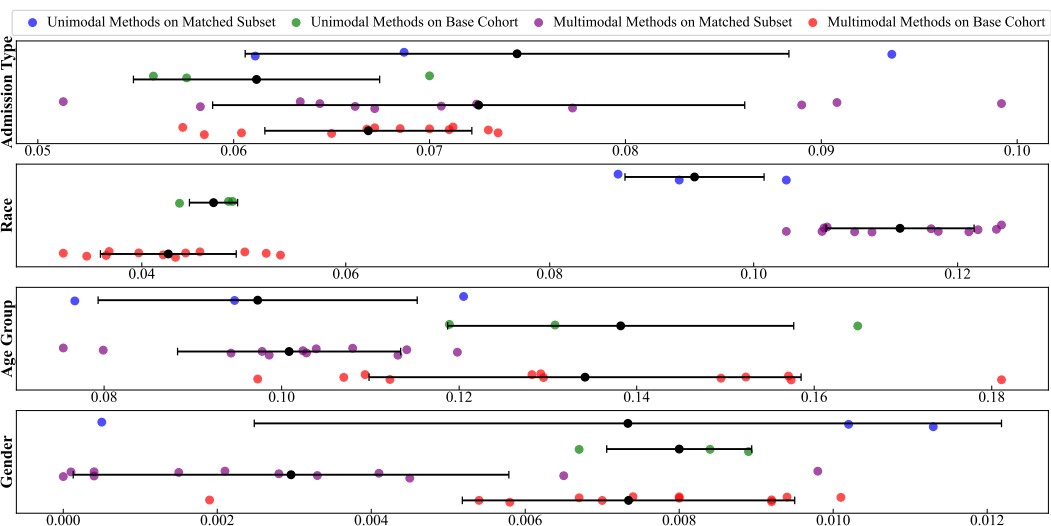

Figure 4: Algorithmic fairness analysis of the AUPRC Gap distributions of unimodal methods and multimodal methods on the matched subset and base cohort.

**Insight 7: Multimodality can exacerbate fairness disparities.** We group the unimodal and multimodal performances in Fig. 4 to analyze the effect of multimodality on fairness. It shows that multimodal methods tend to widen subgroup AUPRC gaps compared to unimodal baselines across age groups and admission types. Multimodal gaps are higher than unimodal gaps for age groups, both on matched subset and base cohort. Similarly, for admission type, multimodal gaps extend up to 0.10, exceeding the 0.06–0.07 range of unimodal methods on matched subset. In contrast, gender disparities remain relatively small across all methods (<0.01), suggesting that fairness degradation is not uniform across attributes. These results indicate that multimodality, while improving overall predictive performance, can amplify disparities between subgroups, especially when data availability or modality completeness is uneven. This underscores the need for fairness-aware multimodal learning strategies that explicitly account for subgroup imbalance and differential modality access.

## 6 CONCLUSION

We introduced CareBench, the first benchmark to evaluate multimodal clinical learners across accuracy, robustness, and fairness using MIMIC-IV and MIMIC-CXR. Our results show that multimodal fusion improves performance when modalities are complete but often fails under high missingness, where unimodal EHR models remain strong baselines. Robustness is highly architecture-dependent, and no single model is optimal across all tasks and metrics. Finally, multimodality can exacerbate fairness disparities, particularly across admission types and age groups, highlighting the need for subgroup-aware and fairness-aware designs in future clinical multimodal systems.

## REPRODUCIBILITY STATEMENT

We ensure reproducibility of our work by releasing the full source code, data preprocessing scripts, and experiment configurations in a public repository (link will be provided upon publication). The repository includes detailed instructions on environment setup, hyperparameter configurations, and data extraction queries to facilitate end-to-end replication of our results. In addition, model checkpoints will be made available upon request to further support reproducibility and enable researchers to validate our reported performance.

## ETHICS STATEMENT

This study was conducted using the Medical Information Mart for Intensive Care (MIMIC) database, a publicly available critical care database hosted on PhysioNet. The database has been de-identified in accordance with the Health Insurance Portability and Accountability Act (HIPAA) standards. Access to the database requires completion of a data use agreement and certification, and our usage fully complies with the guidelines established by the data custodians. No additional patient data were collected, and no individual identities can be inferred from the analyses.

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

## A  DATA EXCLUSION CRITERIA AND DATA STATISTICS

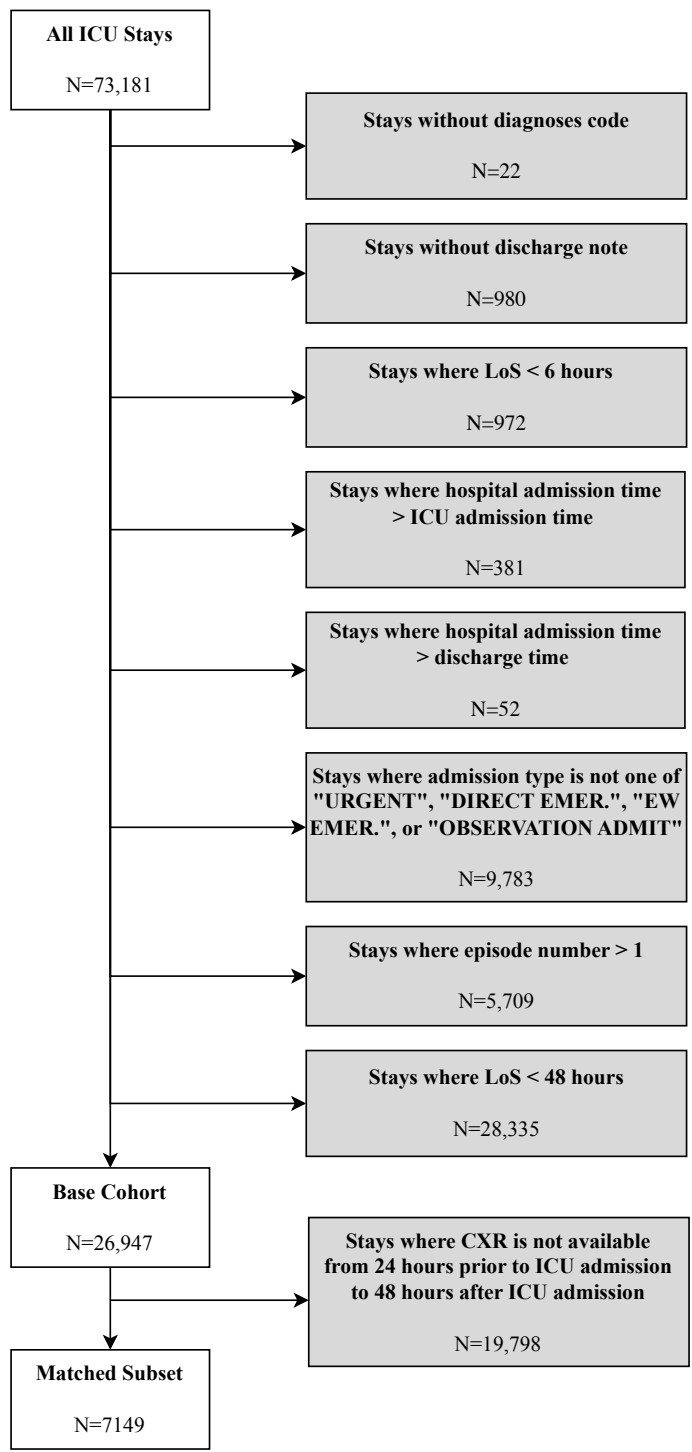

Figure 5: Data exclusion criteria to construct the base cohort and the matched subset.

Fig. 5 summarizes our exclusion criteria in constructing the data cohorts.

We summarize our extracted features in Table 4.

Table 4: Summary of extracted features from MIMIC-IV derived. Continuous variables are reported as mean±std [25%,75%]; categorical variables as median [25%,75%].

| Category | Feature | MIMIC-IV Source | Summary Stats | Missing % |
|---|---|---|---|---|
| **Categorical Features** | | | | |
| Glasgow Coma Scale | Eye Opening | gcs.gcs_eyes | 4 [3,4] | 70.98 |
| | Verbal Response | gcs.gcs_verbal | 5 [0,5] | 71.00 |
| | Motor Response | gcs.gcs_motor | 6 [6,6] | 71.04 |
| | Total Score | gcs.gcs_total | 15 [15,15] | 70.93 |
| Cardiac Rhythm | Absence of Ectopy | mimiciv_icu.chartevents | binary presence | 30.77 |
| | Ectopy type: PVCs | mimiciv_icu.chartevents | binary presence | 89.76 |
| | Atrial Fibrillation (AF) | mimiciv_icu.chartevents | binary presence | 89.91 |
| | Sinus Rhythm (SR) | mimiciv_icu.chartevents | binary presence | 43.89 |
| | Sinus Tachycardia (ST) | mimiciv_icu.chartevents | binary presence | 85.73 |
| **Continuous Features** | | | | |
| Urine Output (KDIGO) | Urine Output | urine_output.urineoutput | 106.44±88.17 [40.00,150.00] | 47.81 |
| | Urine Output rate 6h | kdigo_uo.uo_rt_6hr | 0.93±0.60 [0.48,1.27] | 58.25 |
| | Urine Output rate 12h | kdigo_uo.uo_rt_12hr | 0.92±0.56 [0.50,1.25] | 65.69 |
| | Urine Output rate 24h | kdigo_uo.uo_rt_24hr | 0.93±0.52 [0.53,1.25] | 80.18 |
| | Observation time 6h | kdigo_uo.uo_tm_6hr | 6.95±0.17 [7.00,7.00] | 47.81 |
| | Observation time 12h | kdigo_uo.uo_tm_12hr | 11.68±3.10 [11.00,13.00] | 47.81 |
| | Observation time 24h | kdigo_uo.uo_tm_24hr | 16.77±8.69 [9.00,25.00] | 47.81 |
| Oxygen Delivery | $O_2$ Flow | oxygen_delivery.o2_flow | 3.08±1.43 [2.00,4.00] | 89.48 |
| Ventilator Setting | Fraction of Inspired Oxygen (FiO$_2$) | ventilator_setting.fio2 | 46.93±11.86 [40.00,50.00] | 89.67 |
| Vital Signs | Heart Rate | vitalsign.heart_rate | 84.29±17.34 [72.00,96.00] | 5.19 |
| | Respiratory Rate | vitalsign.resp_rate | 19.12±4.85 [16.00,22.00] | 5.97 |
| | Systolic Blood Pressure | vitalsign.sbp | 118.40±20.21 [103.00,132.00] | 9.04 |
| | Diastolic Blood Pressure | vitalsign.dbp | 62.78±13.61 [53.00,72.00] | 9.06 |
| | Mean Blood Pressure | vitalsign.mbp | 77.76±13.71 [68.00,87.00] | 8.94 |
| | Oxygen Saturation (SpO$_2$) | vitalsign.spo2 | 96.82±2.54 [95.00,99.00] | 7.29 |
| | Temperature | vitalsign.temperature | 36.87±0.48 [36.56,37.17] | 71.89 |
| | Glucose | vitalsign.glucose | 138.36±44.83 [106.00,163.00] | 79.45 |
| Weight | Weight | weight_durations.weight_daily | 81.75±20.28 [66.80,95.00] | 83.78 |

# B  IMPLEMENTATION DETAILS

## B.1  BENCHMARK MODELS

We benchmark a broad set of models for multimodal fusion of EHR and chest X-rays, spanning unimodal baselines, simple fusion strategies, and recent state-of-the-art multimodal algorithms that can be adapted to clinical settings.

**Uni-modal Baselines**    We include uni-modal models as baselines to establish reference performance for each modality. For EHR, we include the classic Long Short-Term Memory network (LSTM) and the Transformer model. They are widely used architectures for capturing temporal dependencies in sequential EHR data. For CXR, we adopt the ResNet-50 model, which is pretrained on ImageNet. These baselines quantify the stand-alone predictive value of each modality.

**Complete-Modality Multimodal Fusion Methods**    This group of models assumes that all modalities are present at both training and inference, including:

- **Unified Temporal Discretization Embedding (UTDE)** (Zhang et al., 2023) is originally designed to handle the irregularity of time series and clinical notes in EHR data. It unifies complementary temporal discretization methods by integrating imputation-based and attention-based interpolation embeddings through a gating mechanism, yielding robust representations of irregular time series. For clinical notes, UTDE casts text embeddings with their note-taking times as irregular sequences and applies a time attention module to capture temporal dynamics. The fusion of time series and notes is then achieved via interleaved self- and cross-attention layers that integrate irregularity across modalities.

- **Dynamic Affine Feature Map Transform (DAFT)** (Pölsterl et al., 2021) is a general-purpose fusion module designed to integrate high-dimensional images with complementary low-dimensional tabular data. DAFT dynamically rescales and shifts convolutional feature maps conditional on tabular inputs, enabling fine-grained interaction between modalities beyond simple concatenation. This mechanism allows clinical variables to modulate intermediate image representations, thereby supporting tighter cross-modal exchange.

- **Multimodal Transfer Module (MMTM)** (Joze et al., 2020) introduces a lightweight plug-in module for CNN-based intermediate fusion. It performs slow fusion by inserting squeeze-and-excitation units into intermediate levels of unimodal backbones, learning a joint representation that adaptively recalibrates channel-wise features across modalities.

- **Late Fusion** is a naive multimodal fusion strategy that concatenates unimodal embeddings followed by a classifier. The EHR and CXR are encoded by a Transformer and a ResNet-50, respectively, with encoders and classifier trained jointly. Despite its simplicity, such late fusion strategies remain widely used in clinical machine learning and serve as strong baselines for comparison against more sophisticated designs.

**Missing-Modality Multimodal Fusion Methods** We implement a broad set of multimodal fusion methods that could handle missing modalities, covering both models developed specifically for clinical data and models originally proposed in other domains (e.g., video–audio classification) that can be naturally adapted to clinical EHR–CXR fusion. This collection spans diverse design paradigms, including:

- **Hybrid Early-fusion Attention Learning Network (HEALNet)** (Hemker et al., 2024) introduces a multimodal fusion architecture that combines shared and modality-specific parameter spaces within an iterative attention framework. A shared latent bottleneck array is propagated and updated across layers to capture cross-modal interactions and shared information. In parallel, modality-specific attention weights are learned and reused across layers, enabling the model to preserve structural information unique to each modality while maintaining efficient fusion through shared parameters.

- **Flexible Mixture-of-Experts (Flex-MoE)** (Yun et al., 2024) is designed to support arbitrary combinations of input modalities without retraining. It constructs a shared latent space where modality-specific encoders map their features, and employs a mixture-of-experts (MoE) fusion layer that dynamically activates experts depending on the available modalities. This enables the model to flexibly integrate any subset of modalities during inference to enhance robustness to missing data and scalability to new modality combinations.

- **DrFuse** (Yao et al., 2024) is a clinical multimodal fusion method proposed for EHR and chest X-ray images. It tackles two key challenges, namely the missing modalities and modal inconsistency. It disentangles shared information (common across EHR and CXR) from modality-specific features and aligns the shared representations via distribution matching. This allows robust inference even when one modality is absent. To further handle patient- and disease-specific variability, DrFuse introduces a disease-aware attention fusion module that adaptively weights each modality. Following its original settings, we adopt Transformer and ResNet-50 as the encoders for EHR and CXR, respectively.

- **Unified Multi-modal Set Embedding (UMSE)** (Lee et al., 2023) addresses the irregular sampling and missing modalities in multimodal EHR learning. It encodes values, time, and feature types across all modalities within a shared embedding framework. By sharing the time and feature embeddings, UMSE preserves temporal relationships between heterogeneous modalities without relying on carry-forward or imputation. To tackle the missing modalities, a Skip Bottleneck (SB) is introduced to enable the Multimodal Bottleneck Transformer to process data with missing modality.

- **Shared-Specific Feature Modelling (ShaSpec)** (Wang et al., 2023) is a multimodal learning framework that decomposes each modality into shared features that are modality-robust and specific features that capture modality-unique information. These components are combined through a residual fusion mechanism. To enforce disentanglement, ShaSpec applies distribution alignment on shared features and a domain classification objective on modality-specific features.

- **M3Care** (Zhang et al., 2022) addresses the challenge of missing modalities in multimodal healthcare data. Instead of generating raw missing data, it imputes task-relevant latent representations by leveraging auxiliary information from clinically similar patients. Specifically, M3Care employs task-guided modality-adaptive kernels to construct patient similarity graphs, aggregates information from these neighbors, and adaptively fuses it with available modalities.

- **MedFuse** (Hayat et al., 2022) is a multimodal fusion method tailored for EHR and chest X-ray images, particularly focusing on missing modalities. After obtaining representations of each modality, MedFuse treats the modality-specific representations (EHR and CXR) as a sequence and aggregates them with an LSTM-based fusion module. This recurrent design enables the model to naturally handle missing modalities by processing variable-length input sequences. Compared to

conventional early or joint fusion, MedFuse improves performance on EHR–CXR prediction tasks while maintaining robustness under partial modality availability.

- **SMIL** (Ma et al., 2021) addresses multimodal learning when a large fraction of training and testing samples lack one or more modalities. It introduces a Bayesian meta-learning framework that contains three components: (i) a feature reconstruction network that approximates missing modality features conditioned on observed ones, (ii) a feature regularization network that perturbs latent embeddings to mitigate bias from incomplete data, and (iii) a main prediction network. This unified design aims to handle different missing-modality patterns during both training and inference, and to train efficiently when most samples are incomplete.

## B.2 FULL MODEL CONFIGURATIONS

Table 5: Overview of model-specific hyperparameter search spaces. General training parameters (learning rate, batch size, epochs, early stopping) are fixed across models.

| Model | Parameters searched | Search space |
|---|---|---|
| DRFuse | $\lambda_{\text{disentangle\_shared}}$, $\lambda_{\text{disentangle\_ehr}}$, $\lambda_{\text{disentangle\_cxr}}$ $\lambda_{\text{pred\_ehr}}$, $\lambda_{\text{pred\_cxr}}$, $\lambda_{\text{pred\_shared}}$, $\lambda_{\text{attn\_aux}}$ | [0.01, 2.0] [0.01, 2.0] |
| FlexMoE | num_experts, num_routers, top-$k$, $\lambda_{\text{gate}}$ | {4, 8, 16}, {1, 2}, {2, 4, 8}, [0.001, 0.1] |
| HealNet | fusion depth, frequency bands, maximum frequency | {1, 2, 3}, {1, 2, 4}, {5.0, 10.0} |
| M3Care | $\lambda_{\text{stab\_reg}}$ | [0.001, 2.0] |
| ShaSpec | $\alpha$ (consistency loss), $\beta$ (domain loss) | [0.01, 0.1], [0.005, 0.2] |
| SMIL | inner loop iters, MC size, inner LR, $\alpha$ (feat distill), $\beta$ (EHR distill), temperature | {1, 2, 3}, {10, 20, 30}, [1e-4, 1e-3], [0.05, 0.2], [0.05, 0.2], [1.0, 3.0] |

Table 6: Full configuration of DRFuse with best hyperparameters from Bayesian optimization acrosstasks and cohorts.

| Category | Parameter | Value / Best value |
|---|---|---|
| General (fixed) | Learning rate | 0.0001 |
| | Batch size | 16 |
| | Epochs | 50 |
| | Patience | 10 |
| | Seeds | {42, 123, 1234} |
| Encoder (fixed) | EHR encoder | Transformer |
| | EHR heads | 4 |
| | EHR layers (distinct/feat/shared) | 1 / 1 / 1 |
| | EHR hidden size | 256 |
| | CXR encoder | ResNet-50 |
| Fusion (fixed) | Fusion method | concatenate |
| | Logit average | true |
| | Attention fusion | mid |
| | Disentangle loss | jsd |
| Phenotype (Base cohort) | $\lambda_{\text{disentangle\_shared}}$ | 0.01 |
| | $\lambda_{\text{disentangle\_ehr}}$ | 0.762665332785317 |
| | $\lambda_{\text{disentangle\_cxr}}$ | 2.0 |
| | $\lambda_{\text{pred\_ehr}}$ | 2.0 |
| | $\lambda_{\text{pred\_cxr}}$ | 2.0 |
| | $\lambda_{\text{pred\_shared}}$ | 2.0 |
| | $\lambda_{\text{attn\_aux}}$ | 1.8578434779578803 |
| Phenotype (Matched subset) | $\lambda_{\text{disentangle\_shared}}$ | 0.47960999030042206 |
| | $\lambda_{\text{disentangle\_ehr}}$ | 0.5195759622950348 |

| Category | Parameter | Value / Best value |
|---|---|---|
| | $\lambda_{\text{disentangle\_cxr}}$ | 0.09046284318147839 |
| | $\lambda_{\text{pred\_ehr}}$ | 1.424219150474717 |
| | $\lambda_{\text{pred\_cxr}}$ | 0.2306727334155444 |
| | $\lambda_{\text{pred\_shared}}$ | 0.8842796387128827 |
| | $\lambda_{\text{attn\_aux}}$ | 0.41142121264743853 |
| Mortality (Base cohort) | $\lambda_{\text{disentangle\_shared}}$ | 0.01 |
| | $\lambda_{\text{disentangle\_ehr}}$ | 0.8112696531851612 |
| | $\lambda_{\text{disentangle\_cxr}}$ | 0.8074771820470718 |
| | $\lambda_{\text{pred\_ehr}}$ | 2.0 |
| | $\lambda_{\text{pred\_cxr}}$ | 1.5418644474046177 |
| | $\lambda_{\text{pred\_shared}}$ | 1.0930962608329169 |
| | $\lambda_{\text{attn\_aux}}$ | 0.016489820579929662 |
| Mortality (Matched subset) | $\lambda_{\text{disentangle\_shared}}$ | 1.854051142929651 |
| | $\lambda_{\text{disentangle\_ehr}}$ | 1.4572712717542777 |
| | $\lambda_{\text{disentangle\_cxr}}$ | 0.6598161299236125 |
| | $\lambda_{\text{pred\_ehr}}$ | 1.145183509066745 |
| | $\lambda_{\text{pred\_cxr}}$ | 1.0464601774513893 |
| | $\lambda_{\text{pred\_shared}}$ | 1.9227323284552051 |
| | $\lambda_{\text{attn\_aux}}$ | 1.6906223588695217 |
| LoS (Base cohort) | $\lambda_{\text{disentangle\_shared}}$ | 0.011549744023618514 |
| | $\lambda_{\text{disentangle\_ehr}}$ | 1.9845010029895234 |
| | $\lambda_{\text{disentangle\_cxr}}$ | 1.238788204159156 |
| | $\lambda_{\text{pred\_ehr}}$ | 1.2271897893716792 |
| | $\lambda_{\text{pred\_cxr}}$ | 0.02406194738723764 |
| | $\lambda_{\text{pred\_shared}}$ | 0.05589422583241736 |
| | $\lambda_{\text{attn\_aux}}$ | 1.0543015739141945 |
| LoS (Matched subset) | $\lambda_{\text{disentangle\_shared}}$ | 1.854051142929651 |
| | $\lambda_{\text{disentangle\_ehr}}$ | 1.4572712717542777 |
| | $\lambda_{\text{disentangle\_cxr}}$ | 0.6598161299236125 |
| | $\lambda_{\text{pred\_ehr}}$ | 1.145183509066745 |
| | $\lambda_{\text{pred\_cxr}}$ | 1.0464601774513893 |
| | $\lambda_{\text{pred\_shared}}$ | 1.9227323284552051 |
| | $\lambda_{\text{attn\_aux}}$ | 1.6906223588695217 |

Table 7: Full configuration of HealNet with best hyperparameters from Bayesian optimization across tasks and cohorts.

| Category | Parameter | Value / Best value |
|---|---|---|
| General (fixed) | Learning rate | 0.0001 |
| | Batch size | 16 |
| | Epochs | 50 |
| | Patience | 10 |
| | Dropout | 0.2 |
| | Seeds | {42, 123, 1234} |
| Encoder (fixed) | N_modalities | 2 (EHR + CXR) |
| | Latent channels | 256 |
| | Latent dimension | 256 |
| | Cross-attention heads | 4 |
| | Latent attention heads | 4 |
| | Cross head dimension | 64 |
| | Latent head dimension | 64 |
| | Self per cross attention | 1 |
| | Weight tie layers | true |

| Category | Parameter | Value / Best value |
|---|---|---|
| | Self-normalizing nets | true |
| | Fourier encoding | true |
| | Final classifier head | true |
| | Attention dropout | 0.2 |
| | Feed-forward dropout | 0.2 |
| Phenotype (Base cohort) | Fusion depth | 1 |
| | Num frequency bands | 4 |
| | Max frequency | 10 |
| Phenotype (Matched subset) | Fusion depth | 1 |
| | Num frequency bands | 4 |
| | Max frequency | 5 |
| Mortality (Base cohort) | Fusion depth | 1 |
| | Num frequency bands | 4 |
| | Max frequency | 5 |
| Mortality (Matched subset) | Fusion depth | 3 |
| | Num frequency bands | 2 |
| | Max frequency | 5 |
| LoS (Base cohort) | Fusion depth | 3 |
| | Num frequency bands | 1 |
| | Max frequency | 10 |
| LoS (Matched subset) | Fusion depth | 2 |
| | Num frequency bands | 2 |
| | Max frequency | 5 |

Table 8: Full configuration of FlexMoE with best hyperparameters from Bayesian optimization across tasks and cohorts.

| Category | Parameter | Value / Best value |
|---|---|---|
| General (fixed) | Learning rate | 0.0001 |
| | Batch size | 16 |
| | Epochs | 50 |
| | Patience | 10 |
| | Dropout | 0.2 |
| | Seeds | {42, 123, 1234} |
| Encoder (fixed) | EHR encoder | Transformer |
| | CXR encoder | ResNet-50 |
| Architecture (fixed) | Hidden dimension | 256 |
| | Num patches | 16 |
| | Num layers | 1 |
| | Num prediction layers | 1 |
| | Num heads | 4 |
| EHR Transformer (fixed) | Attention heads | 4 |
| | Layers | 1 |
| Phenotype (Base cohort) | Num experts | 8 |
| | Num routers | 2 |
| | Top-$k$ | 4 |
| | Gate loss weight | 0.1 |
| Phenotype (Matched subset) | Num experts | 8 |
| | Num routers | 2 |
| | Top-$k$ | 4 |

| Category | Parameter | Value / Best value |
|---|---|---|
| | Gate loss weight | 0.01 |
| Mortality (Base cohort) | Num experts | 16 |
| | Num routers | 1 |
| | Top-$k$ | 2 |
| | Gate loss weight | 0.001 |
| Mortality (Matched subset) | Num experts | 16 |
| | Num routers | 1 |
| | Top-$k$ | 8 |
| | Gate loss weight | 0.059157941793120575 |
| LoS (Base cohort) | Num experts | 8 |
| | Num routers | 2 |
| | Top-$k$ | 2 |
| | Gate loss weight | 0.09949248941642917 |
| LoS (Matched subset) | Num experts | 4 |
| | Num routers | 2 |
| | Top-$k$ | 2 |
| | Gate loss weight | 0.001 |

Table 9: Full configuration of M3Care with best hyperparameters from Bayesian optimization across tasks and cohorts.

| Category | Parameter | Value / Best value |
|---|---|---|
| General (fixed) | Learning rate | 0.0001 |
| | Batch size | 16 |
| | Epochs | 50 |
| | Patience | 10 |
| | Dropout | 0.2 |
| | Seeds | {42, 123, 1234} |
| Encoder (fixed) | EHR encoder | Transformer |
| | CXR encoder | ResNet-50 |
| Architecture (fixed) | Hidden dimension | 256 |
| | EHR attention heads | 4 |
| | EHR layers | 1 |
| | Max sequence length | 500 |
| | LSTM bidirectional | true |
| | LSTM layers | 1 |
| Search (M3Care-specific) | $\lambda_{\text{stab\_reg}}$ | [0.001, 2.0] |
| Phenotype (Base cohort) | $\lambda_{\text{stab\_reg}}$ | 0.001 |
| Phenotype (Matched subset) | $\lambda_{\text{stab\_reg}}$ | 0.001 |
| Mortality (Base cohort) | $\lambda_{\text{stab\_reg}}$ | 0.001 |
| Mortality (Matched subset) | $\lambda_{\text{stab\_reg}}$ | 1.5932894307336058 |
| LoS (Base cohort) | $\lambda_{\text{stab\_reg}}$ | 0.1865396435927205 |
| LoS (Matched subset) | $\lambda_{\text{stab\_reg}}$ | 0.7189374969280278 |

Table 10: Full configuration of ShaSpec with best hyperparameters from Bayesian optimization across tasks and cohorts.

| Category | Parameter | Value / Best value |
|---|---|---|
| General (fixed) | Learning rate | 0.0001 |

| Category | Parameter | Value / Best value |
|---|---|---|
| | Batch size | 16 |
| | Epochs | 50 |
| | Patience | 10 |
| | Dropout | 0.2 |
| | Seeds | {42, 123, 1234} |
| Encoder (fixed) | EHR encoder | Transformer |
| | CXR encoder | ResNet-50 |
| Architecture (fixed) | Hidden dimension | 256 |
| | Weight standardization | true |
| | EHR attention heads | 4 |
| | EHR layers | 1 |
| | Shared transformer heads | 4 |
| | Shared transformer layers | 1 |
| | Max sequence length | 500 |
| Phenotype (Base cohort) | $\alpha$ | 0.01 |
| | $\beta$ | 0.16088171387061506 |
| Phenotype (Matched subset) | $\alpha$ | 0.01 |
| | $\beta$ | 0.02179517168618064 |
| Mortality (Base cohort) | $\alpha$ | 0.02616340058300993 |
| | $\beta$ | 0.0283073339627612397 |
| Mortality (Matched subset) | $\alpha$ | 0.05309286239046574 |
| | $\beta$ | 0.05759057147063285 |
| LoS (Base cohort) | $\alpha$ | 0.0539365721371763 |
| | $\beta$ | 0.02400870516077973 |
| LoS (Matched subset) | $\alpha$ | 0.08168886881742098 |
| | $\beta$ | 0.040769784023901946 |

Table 11: Full configuration of SMIL with best hyperparameters from Bayesian optimization across tasks and cohorts.

| Category | Parameter | Value / Best value |
|---|---|---|
| General (fixed) | Learning rate | 0.0001 |
| | Batch size | 16 |
| | Epochs | 50 |
| | Patience | 10 |
| | Dropout | 0.2 |
| | Seeds | {42, 123, 1234} |
| Encoder (fixed) | EHR encoder | Transformer |
| | CXR encoder | ResNet-50 |
| Architecture (fixed) | Hidden dimension | 256 |
| | EHR attention heads | 4 |
| | EHR layers | 1 |
| | Max sequence length | 500 |
| | Number of clusters | 10 |
| Phenotype (Base cohort) | Inner loop iterations | 2 |
| | Monte Carlo size | 20 |
| | Inner learning rate | 0.0007631400164186543 |
| | $\alpha$ | 0.05 |
| | $\beta$ | 0.08821668183456577 |
| | Temperature | 3.0 |

| Category | Parameter | Value / Best value |
|---|---|---|
| Phenotype (Matched subset) | Inner loop iterations | 1 |
| | Monte Carlo size | 20 |
| | Inner learning rate | 0.000812341589273 |
| | $\alpha$ | 0.052 |
| | $\beta$ | 0.091 |
| | Temperature | 2.647382910384756 |
| Mortality (Base cohort) | Inner loop iterations | 2 |
| | Monte Carlo size | 10 |
| | Inner learning rate | 0.0005106366481617694 |
| | $\alpha$ | 0.14609708292122307 |
| | $\beta$ | 0.1829896904844842 |
| | Temperature | 2.1659838411477903 |
| Mortality (Matched subset) | Inner loop iterations | 2 |
| | Monte Carlo size | 20 |
| | Inner learning rate | 0.000498272163782 |
| | $\alpha$ | 0.141 |
| | $\beta$ | 0.176 |
| | Temperature | 2.223746192837465 |
| LoS (Base cohort) | Inner loop iterations | 3 |
| | Monte Carlo size | 20 |
| | Inner learning rate | 0.00025110824439271776 |
| | $\alpha$ | 0.05699984948204232 |
| | $\beta$ | 0.19606332782621894 |
| | Temperature | 1.4655426808606085 |
| LoS (Matched subset) | Inner loop iterations | 3 |
| | Monte Carlo size | 10 |
| | Inner learning rate | 0.00048099463193341493 |
| | $\alpha$ | 0.10998628646125844 |
| | $\beta$ | 0.17555899317323298 |
| | Temperature | 1.0842555683358894 |

Table 12: Full configuration of DAFT. No hyperparameter search was performed as all parameters are fixed.

| Category | Parameter | Value |
|---|---|---|
| General (fixed) | Learning rate | 0.0001 |
| | Batch size | 16 |
| | Epochs | 50 |
| | Patience | 10 |
| | Seeds | {42, 123, 1234} |
| | Dropout | 0.2 |
| Encoder (fixed) | EHR encoder | Transformer |
| | CXR encoder | ResNet-50 |
| | EHR attention heads | 4 |
| | EHR layers | 1 |
| DAFT fusion (fixed) | Layer after | -1 (all layers) |
| | Activation | linear |
| Architecture (fixed) | Hidden dimension | 256 |

Table 13: Full configuration of LateFusion. No hyperparameter search was performed as all parameters are fixed.

| Category | Parameter | Value |
|---|---|---|
| General (fixed) | Learning rate | 0.0001 |
| | Batch size | 16 |
| | Epochs | 50 |
| | Patience | 10 |
| | Dropout | 0.2 |
| | Seeds | {42, 123, 1234} |
| Encoder (fixed) | EHR encoder | Transformer |
| | CXR encoder | ResNet-50 |
| Architecture (fixed) | Hidden size | 256 |
| | EHR layers | 1 |
| | EHR attention heads | 4 |
| | EHR dropout | 0.2 |

Table 14: Full configuration of the LSTM. No hyperparameter search was performed as all parameters are fixed.

| Category | Parameter | Value |
|---|---|---|
| General (fixed) | Learning rate | 0.0001 |
| | Batch size | 16 |
| | Epochs | 50 |
| | Patience | 10 |
| | Dropout | 0.2 |
| | Seeds | {42, 123, 1234} |
| Architecture (fixed) | Hidden size | 256 |
| | Num layers | 1 |
| | Bidirectional | true |
| | Dropout | 0.2 |

Table 15: Full configuration of MedFuse. No hyperparameter search was performed as all parameters are fixed.

| Category | Parameter | Value |
|---|---|---|
| General (fixed) | Learning rate | 0.0001 |
| | Batch size | 16 |
| | Epochs | 50 |
| | Patience | 10 |
| | Dropout | 0.2 |
| | Seeds | {42, 123, 1234} |
| Encoder (fixed) | EHR encoder | LSTM |
| | CXR encoder | ResNet-50 |
| | EHR LSTM bidirectional | true |
| Architecture (fixed) | Hidden dimension | 256 |
| | LSTM layers | 1 |
| | Fusion type | LSTM |

Table 16: Full configuration of MMTM. No hyperparameter search was performed as all parameters are fixed.

| Category | Parameter | Value |
|---|---|---|
| General (fixed) | Learning rate | 0.0001 |
| | Batch size | 16 |
| | Epochs | 50 |
| | Patience | 10 |
| | Dropout | 0.2 |
| | Seeds | {42, 123, 1234} |
| Encoder (fixed) | EHR encoder | Transformer |
| | CXR encoder | ResNet-50 |
| Architecture (fixed) | Hidden dimension | 256 |
| | EHR attention heads | 4 |
| | EHR layers | 1 |
| MMTM Fusion (fixed) | Compression ratio | 4 |
| | Layer after | -1 (all layers) |

Table 17: Full configuration of the ResNet. No hyperparameter search was performed as all parameters are fixed.

| Category | Parameter | Value |
|---|---|---|
| General (fixed) | Learning rate | 0.0001 |
| | Batch size | 16 |
| | Epochs | 50 |
| | Patience | 10 |
| | Dropout | 0.2 |
| | Seeds | {42, 123, 1234} |
| Architecture (fixed) | Hidden size | 256 |

Table 18: Full configuration of UMSE. No hyperparameter search was performed as all parameters are fixed.

| Category | Parameter | Value |
|---|---|---|
| General (fixed) | Learning rate | 0.0001 |
| | Batch size | 16 |
| | Epochs | 50 |
| | Patience | 10 |
| | Dropout | 0.2 |
| | Seeds | {42, 123, 1234} |
| Architecture (fixed) | Model dimension | 256 |
| | Transformer layers | 1 |
| | Attention heads | 4 |
| Fusion (fixed) | Bottlenecks (MBT) | 1 |

Table 19: Full configuration of UTDE. No hyperparameter search was performed as all parameters are fixed.

| Category | Parameter | Value |
|---|---|---|
| General (fixed) | Learning rate | 0.0001 |
| | Batch size | 16 |
| | Epochs | 50 |
| | Patience | 10 |
| | Dropout | 0.2 |
| | Seeds | {42, 123, 1234} |
| Encoder (fixed) | EHR encoder | Transformer |
| | CXR encoder | ResNet-50 |
| Architecture (fixed) | Embedding dimension | 256 |
| | EHR num layers | 1 |
| | EHR attention heads | 4 |
| | Time embedding dimension | 64 |
| | Transformer attention heads | 4 |
| | Cross-modal layers | 1 |
| | Max EHR sequence length | 500 |

Table 20: Full configuration of the Transformer baseline. No hyperparameter search was performed as all parameters are fixed.

| Category | Parameter | Value |
|---|---|---|
| General (fixed) | Learning rate | 0.0001 |
| | Batch size | 16 |
| | Epochs | 50 |
| | Patience | 10 |
| | Dropout | 0.2 |
| | Seeds | {42, 123, 1234} |
| Architecture (fixed) | Model dimension | 256 |
| | Transformer layers | 1 |
| | Attention heads | 4 |

## C COMPLETE EXPERIMENT RESULTS

### C.1 OVERVIEW RESULTS

We provide the complete experiment results in the tables below.

Table 21: Complete results of phenotype classification task on the matched subset.

| | AUROC | AUPRC | F1 | Precision | Recall | Specificity | ACC |
|---|---|---|---|---|---|---|---|
| **LSTM** | 0.7153±0.0057 | 0.4520±0.0036 | 0.3279±0.0227 | 0.4978±0.0065 | 0.2792±0.0257 | 0.8998±0.0113 | 0.7801±0.0011 |
| **Transformer** | 0.7244±0.0005 | 0.4718±0.0008 | 0.3395±0.0123 | 0.5097±0.0156 | 0.2845±0.0183 | 0.9037±0.0086 | 0.7842±0.0004 |
| **ResNet** | 0.6679±0.0022 | 0.3997±0.0014 | 0.2693±0.0145 | 0.4573±0.0354 | 0.2261±0.0200 | 0.9018±0.0136 | 0.7665±0.0019 |
| **UTDE** | 0.7402±0.0015 | 0.4920±0.0036 | 0.3029±0.0198 | 0.5540±0.0202 | 0.2262±0.0210 | 0.9493±0.0101 | 0.7858±0.0026 |
| **DAFT** | 0.7307±0.0034 | 0.4799±0.0034 | 0.2750±0.0217 | 0.5560±0.0073 | 0.2039±0.0205 | 0.9512±0.0072 | 0.7830±0.0014 |
| **MMTM** | 0.7269±0.0012 | 0.4723±0.0048 | 0.3808±0.0116 | 0.5134±0.0125 | 0.3319±0.0263 | 0.8891±0.0123 | 0.7800±0.0026 |
| **LateFusion** | 0.7381±0.0012 | 0.4890±0.0007 | 0.2977±0.0182 | 0.5612±0.0074 | 0.2231±0.0201 | 0.9476±0.0094 | 0.7860±0.0006 |
| **HEALNet** | 0.7261±0.0012 | 0.4714±0.0009 | 0.2104±0.0186 | 0.5657±0.0135 | 0.1441±0.0150 | 0.9671±0.0059 | 0.7792±0.0009 |
| **Flex-MoE** | 0.7355±0.0007 | 0.4876±0.0022 | 0.2968±0.0205 | 0.5643±0.0301 | 0.2211±0.0184 | 0.9457±0.0041 | 0.7850±0.0025 |
| **DrFuse** | 0.7387±0.0008 | 0.4928±0.0024 | 0.3762±0.0196 | 0.5382±0.0173 | 0.3284±0.0269 | 0.8971±0.0093 | 0.7884±0.0021 |
| **UMSE** | 0.7106±0.0018 | 0.4462±0.0016 | 0.1916±0.0026 | 0.4879±0.0419 | 0.1355±0.0016 | 0.9618±0.0031 | 0.7748±0.0004 |
| **ShaSpec** | 0.7342±0.0017 | 0.4813±0.0046 | 0.3414±0.0226 | 0.4946±0.0082 | 0.2858±0.0270 | 0.9150±0.0124 | 0.7883±0.0024 |
| **M3Care** | 0.7388±0.0014 | 0.4881±0.0026 | 0.2750±0.0251 | 0.5349±0.0091 | 0.2048±0.0246 | 0.9511±0.0071 | 0.7842±0.0018 |
| **MedFuse** | 0.7317±0.0014 | 0.4777±0.0030 | 0.2556±0.0191 | 0.4972±0.0272 | 0.1891±0.0194 | 0.9545±0.0059 | 0.7823±0.0011 |
| **SMIL** | 0.7130±0.0034 | 0.4517±0.0034 | 0.3081±0.0109 | 0.4906±0.0375 | 0.2642±0.0136 | 0.9050±0.0092 | 0.7813±0.0017 |

Table 22: Complete results of phenotype classification task on the base cohort.

| | AUROC | AUPRC | F1 | Precision | Recall | Specificity | ACC |
|---|---|---|---|---|---|---|---|
| **LSTM** | 0.7547±0.0003 | 0.4684±0.0013 | 0.3336±0.0224 | 0.5506±0.0113 | 0.2729±0.0209 | 0.9250±0.0070 | 0.8055±0.0003 |
| **Transformer** | 0.7591±0.0003 | 0.4787±0.0011 | 0.3618±0.0041 | 0.5630±0.0156 | 0.2972±0.0043 | 0.9218±0.0013 | 0.8075±0.0007 |
| **ResNet** | 0.5650±0.0001 | 0.2801±0.0021 | 0.1149±0.0104 | 0.4474±0.0331 | 0.0700±0.0069 | 0.9745±0.0030 | 0.7791±0.0003 |
| **UTDE** | 0.7636±0.0004 | 0.4853±0.0011 | 0.2787±0.0105 | 0.6101±0.0235 | 0.2005±0.0101 | 0.9598±0.0012 | 0.8060±0.0005 |
| **DAFT** | 0.7565±0.0007 | 0.4755±0.0025 | 0.2351±0.0192 | 0.6284±0.0160 | 0.1594±0.0175 | 0.9701±0.0046 | 0.8017±0.0016 |
| **MMTM** | 0.7553±0.0003 | 0.4705±0.0009 | 0.3390±0.0145 | 0.5483±0.0120 | 0.2795±0.0186 | 0.9200±0.0078 | 0.8049±0.0007 |
| **LateFusion** | 0.7625±0.0017 | 0.4847±0.0006 | 0.2891±0.0163 | 0.6282±0.0244 | 0.2064±0.0165 | 0.9612±0.0059 | 0.8057±0.0005 |
| **HEALNet** | 0.7578±0.0005 | 0.4752±0.0008 | 0.2196±0.0041 | 0.6349±0.0211 | 0.1474±0.0035 | 0.9734±0.0006 | 0.8009±0.0003 |
| **Flex-MoE** | 0.7619±0.0016 | 0.4835±0.0023 | 0.3162±0.0084 | 0.5790±0.0102 | 0.2335±0.0090 | 0.9521±0.0042 | 0.8068±0.0002 |
| **DrFuse** | 0.7639±0.0005 | 0.4845±0.0004 | 0.3613±0.0048 | 0.5683±0.0069 | 0.2925±0.0082 | 0.9266±0.0054 | 0.8086±0.0004 |
| **UMSE** | 0.7482±0.0015 | 0.4564±0.0028 | 0.1733±0.0015 | 0.6079±0.0068 | 0.1119±0.0013 | 0.9788±0.0010 | 0.7952±0.0002 |
| **ShaSpec** | 0.7626±0.0003 | 0.4848±0.0009 | 0.3473±0.0139 | 0.5671±0.0101 | 0.2833±0.0180 | 0.9261±0.0072 | 0.8084±0.0005 |
| **M3Care** | 0.7637±0.0007 | 0.4883±0.0008 | 0.2713±0.0109 | 0.6375±0.0269 | 0.1935±0.0084 | 0.9623±0.0019 | 0.8063±0.0003 |
| **MedFuse** | 0.7556±0.0009 | 0.4686±0.0016 | 0.2204±0.0003 | 0.5622±0.0212 | 0.1534±0.0006 | 0.9692±0.0004 | 0.8020±0.0005 |
| **SMIL** | 0.7420±0.0007 | 0.4474±0.0004 | 0.2964±0.0063 | 0.4802±0.0108 | 0.2469±0.0083 | 0.9267±0.0038 | 0.8032±0.0006 |

Table 23: Complete results of mortality prediction task on the matched subset.

| | AUROC | AUPRC | F1 | Precision | Recall | Specificity | ACC |
|---|---|---|---|---|---|---|---|
| **LSTM** | 0.8032±0.0261 | 0.4509±0.0243 | 0.3192±0.1220 | 0.5672±0.0942 | 0.2549±0.1235 | 0.9674±0.0200 | 0.8820±0.0029 |
| **Transformer** | 0.8208±0.0036 | 0.4474±0.0151 | 0.2131±0.0700 | 0.7501±0.1597 | 0.1333±0.0601 | 0.9901±0.0107 | 0.8874±0.0022 |
| **ResNet** | 0.6879±0.0090 | 0.2252±0.0062 | 0.0495±0.0479 | 0.2667±0.2055 | 0.0294±0.0300 | 0.9917±0.0101 | 0.8764±0.0053 |
| **UTDE** | 0.8228±0.0118 | 0.4510±0.0120 | 0.2791±0.1810 | 0.6841±0.2235 | 0.2255±0.1516 | 0.9728±0.0194 | 0.8832±0.0014 |
| **DAFT** | 0.8342±0.0050 | 0.4354±0.0131 | 0.0078±0.0110 | 0.3333±0.4714 | 0.0039±0.0055 | 1.0000±0.0000 | 0.8806±0.0007 |
| **MMTM** | 0.8007±0.0323 | 0.3571±0.0540 | 0.3201±0.1181 | 0.3818±0.0460 | 0.3020±0.1430 | 0.9364±0.0283 | 0.8604±0.0110 |
| **LateFusion** | 0.8233±0.0039 | 0.4328±0.0179 | 0.2871±0.0924 | 0.5707±0.0738 | 0.2137±0.1066 | 0.9736±0.0207 | 0.8825±0.0057 |
| **HEALNet** | 0.8356±0.0039 | 0.4507±0.0074 | 0.1752±0.0400 | 0.7555±0.0227 | 0.1000±0.0254 | 0.9955±0.0015 | 0.8881±0.0017 |
| **Flex-MoE** | 0.8401±0.0054 | 0.4734±0.0230 | 0.2664±0.1722 | 0.7018±0.2205 | 0.2176±0.1658 | 0.9730±0.0271 | 0.8825±0.0066 |
| **DrFuse** | 0.8378±0.0034 | 0.4813±0.0091 | 0.3375±0.0695 | 0.5809±0.0853 | 0.2569±0.0933 | 0.9704±0.0191 | 0.8848±0.0056 |
| **UMSE** | 0.8014±0.0077 | 0.3949±0.0145 | 0.0039±0.0055 | 0.3333±0.4714 | 0.0020±0.0028 | 0.9995±0.0008 | 0.8799±0.0009 |
| **ShaSpec** | 0.8331±0.0035 | 0.4527±0.0006 | 0.2204±0.0505 | 0.7028±0.0491 | 0.1333±0.0391 | 0.9917±0.0043 | 0.8888±0.0012 |
| **M3Care** | 0.8277±0.0008 | 0.4447±0.0114 | 0.2410±0.0506 | 0.6910±0.1501 | 0.1530±0.0458 | 0.9875±0.0095 | 0.8874±0.0029 |
| **MedFuse** | 0.8509±0.0056 | 0.4717±0.0098 | 0.1569±0.0892 | 0.7968±0.1473 | 0.0922±0.0546 | 0.9949±0.0036 | 0.8867±0.0037 |
| **SMIL** | 0.8372±0.0026 | 0.4532±0.0200 | 0.2641±0.0250 | 0.6848±0.0413 | 0.1647±0.0210 | 0.9893±0.0031 | 0.8905±0.0003 |

Table 25: Complete results of the length of stay prediction task on the matched subset.

| | F1 | Precision | Recall | Specificity | ACC | Kappa |
|---|---|---|---|---|---|---|
| **LSTM** | 0.1864±0.0136 | 0.2016±0.0205 | 0.2336±0.0062 | 0.8852±0.0010 | 0.3973±0.0044 | 0.1925±0.0060 |
| **Transformer** | 0.1923±0.0070 | 0.2091±0.0098 | 0.2336±0.0034 | 0.8844±0.0015 | 0.3921±0.0101 | 0.1870±0.0106 |
| **ResNet** | 0.1333±0.0025 | 0.1296±0.0083 | 0.1942±0.0007 | 0.8737±0.0002 | 0.3463±0.0076 | 0.1151±0.0022 |
| **UTDE** | 0.1930±0.0063 | 0.2029±0.0126 | 0.2333±0.0007 | 0.8857±0.0007 | 0.3945±0.0066 | 0.1946±0.0056 |
| **DAFT** | 0.1604±0.0133 | 0.1625±0.0395 | 0.2224±0.0043 | 0.8832±0.0014 | 0.3937±0.0030 | 0.1800±0.0087 |
| **MMTM** | 0.1648±0.0069 | 0.1825±0.0193 | 0.2204±0.0119 | 0.8704±0.0011 | 0.2607±0.0062 | 0.0907±0.0089 |
| **LateFusion** | 0.1886±0.0094 | 0.2659±0.0218 | 0.2360±0.0021 | 0.8849±0.0006 | 0.3940±0.0068 | 0.1910±0.0053 |
| **HEALNet** | 0.1866±0.0119 | 0.2426±0.0372 | 0.2348±0.0058 | 0.8853±0.0003 | 0.4036±0.0013 | 0.1948±0.0027 |
| **Flex-MoE** | 0.1870±0.0056 | 0.2160±0.0333 | 0.2337±0.0027 | 0.8855±0.0003 | 0.3980±0.0022 | 0.1946±0.0014 |
| **DrFuse** | 0.1847±0.0123 | 0.2253±0.0708 | 0.2347±0.0066 | 0.8850±0.0003 | 0.4060±0.0067 | 0.1947±0.0046 |
| **UMSE** | 0.1602±0.0142 | 0.1572±0.0224 | 0.2224±0.0054 | 0.8840±0.0009 | 0.3912±0.0076 | 0.1826±0.0075 |
| **ShaSpec** | 0.1936±0.0082 | 0.1954±0.0231 | 0.2354±0.0054 | 0.8861±0.0010 | 0.3980±0.0053 | 0.1978±0.0074 |
| **M3Care** | 0.1602±0.0067 | 0.1913±0.0448 | 0.2238±0.0043 | 0.8839±0.0010 | 0.4003±0.0063 | 0.1854±0.0077 |
| **MedFuse** | 0.1672±0.0078 | 0.1922±0.0710 | 0.2273±0.0022 | 0.8854±0.0005 | 0.4031±0.0013 | 0.1942±0.0035 |
| **SMIL** | 0.1460±0.0023 | 0.1479±0.0302 | 0.2191±0.0024 | 0.8830±0.0009 | 0.3989±0.0027 | 0.1791±0.0054 |

Table 26: Complete results of the length of stay prediction task on the base cohort.

| | F1 | Precision | Recall | Specificity | ACC | Kappa |
|---|---|---|---|---|---|---|
| **LSTM** | 0.1781±0.0043 | 0.1919±0.0199 | 0.2313±0.0023 | 0.8859±0.0005 | 0.4120±0.0034 | 0.1966±0.0034 |
| **Transformer** | 0.1976±0.0144 | 0.2232±0.0049 | 0.2425±0.0084 | 0.8869±0.0010 | 0.4171±0.0033 | 0.2040±0.0053 |
| **ResNet** | 0.1059±0.0008 | 0.1466±0.0628 | 0.1608±0.0008 | 0.8625±0.0002 | 0.3542±0.0005 | 0.0408±0.0018 |
| **UTDE** | 0.2013±0.0057 | 0.2209±0.0226 | 0.2449±0.0030 | 0.8874±0.0000 | 0.4162±0.0006 | 0.2071±0.0002 |
| **DAFT** | 0.1977±0.0106 | 0.2196±0.0373 | 0.2435±0.0041 | 0.8876±0.0005 | 0.4179±0.0007 | 0.2080±0.0031 |
| **MMTM** | 0.1615±0.0118 | 0.1669±0.0154 | 0.2205±0.0012 | 0.8817±0.0005 | 0.4039±0.0020 | 0.1714±0.0032 |
| **LateFusion** | 0.2001±0.0096 | 0.2134±0.0214 | 0.2451±0.0041 | 0.8880±0.0009 | 0.4161±0.0011 | 0.2101±0.0046 |
| **HEALNet** | 0.1906±0.0117 | 0.2508±0.0322 | 0.2374±0.0057 | 0.8861±0.0011 | 0.4182±0.0011 | 0.1998±0.0062 |
| **Flex-MoE** | 0.1879±0.0092 | 0.1943±0.0049 | 0.2406±0.0031 | 0.8875±0.0008 | 0.4137±0.0025 | 0.2059±0.0038 |
| **DrFuse** | 0.1906±0.0148 | 0.2376±0.0116 | 0.2381±0.0085 | 0.8854±0.0008 | 0.4190±0.0015 | 0.1965±0.0051 |
| **UMSE** | 0.1814±0.0041 | 0.1819±0.0035 | 0.2301±0.0035 | 0.8839±0.0009 | 0.4054±0.0027 | 0.1842±0.0056 |
| **ShaSpec** | 0.1912±0.0036 | 0.1961±0.0022 | 0.2413±0.0012 | 0.8876±0.0003 | 0.4208±0.0008 | 0.2088±0.0013 |
| **M3Care** | 0.1879±0.0110 | 0.1935±0.0020 | 0.2386±0.0042 | 0.8869±0.0006 | 0.4195±0.0016 | 0.2044±0.0031 |
| **MedFuse** | 0.2035±0.0085 | 0.2185±0.0190 | 0.2467±0.0045 | 0.8883±0.0005 | 0.4177±0.0028 | 0.2127±0.0028 |
| **SMIL** | 0.1570±0.0059 | 0.1432±0.0046 | 0.2269±0.0028 | 0.8849±0.0009 | 0.4171±0.0020 | 0.1921±0.0051 |

Table 24: Complete results of mortality prediction task on the base cohort.

| | AUROC | AUPRC | F1 | Precision | Recall | Specificity | ACC |
|---|---|---|---|---|---|---|---|
| **LSTM** | 0.8608±0.0061 | 0.4797±0.0131 | 0.3668±0.0459 | 0.6403±0.0323 | 0.2602±0.0455 | 0.9843±0.0043 | 0.9151±0.0023 |
| **Transformer** | 0.8674±0.0010 | 0.5042±0.0059 | 0.4024±0.0277 | 0.6417±0.0469 | 0.2971±0.0412 | 0.9817±0.0063 | 0.9163±0.0020 |
| **ResNet** | 0.5655±0.0143 | 0.1339±0.0086 | 0.0013±0.0018 | 0.1667±0.2357 | 0.0006±0.0009 | 0.9999±0.0001 | 0.9045±0.0000 |
| **UTDE** | 0.8683±0.0015 | 0.5029±0.0048 | 0.2864±0.0605 | 0.7739±0.0355 | 0.1786±0.0454 | 0.9942±0.0023 | 0.9163±0.0024 |
| **DAFT** | 0.8698±0.0019 | 0.4952±0.0057 | 0.2297±0.0791 | 0.7949±0.0616 | 0.1392±0.0572 | 0.9955±0.0033 | 0.9137±0.0026 |
| **MMTM** | 0.8649±0.0004 | 0.4749±0.0031 | 0.3720±0.0425 | 0.6277±0.0503 | 0.2699±0.0495 | 0.9822±0.0066 | 0.9141±0.0019 |
| **LateFusion** | 0.8672±0.0023 | 0.4995±0.0067 | 0.3386±0.0384 | 0.7390±0.0433 | 0.2220±0.0355 | 0.9913±0.0032 | 0.9178±0.0009 |
| **HEALNet** | 0.8729±0.0018 | 0.4914±0.0055 | 0.2911±0.0830 | 0.7288±0.0381 | 0.1890±0.0722 | 0.9919±0.0047 | 0.9152±0.0027 |
| **Flex-MoE** | 0.8684±0.0014 | 0.5065±0.0061 | 0.3054±0.1041 | 0.7451±0.1044 | 0.2071±0.0914 | 0.9900±0.0086 | 0.9152±0.0030 |
| **DrFuse** | 0.8736±0.0035 | 0.4999±0.0078 | 0.3561±0.0172 | 0.6784±0.0206 | 0.2421±0.0184 | 0.9878±0.0021 | 0.9165±0.0005 |
| **UMSE** | 0.8455±0.0053 | 0.4337±0.0047 | 0.1507±0.0457 | 0.7413±0.0593 | 0.0854±0.0296 | 0.9964±0.0021 | 0.9094±0.0009 |
| **ShaSpec** | 0.8690±0.0027 | 0.5000±0.0068 | 0.3882±0.0433 | 0.6383±0.0447 | 0.2841±0.0502 | 0.9822±0.0060 | 0.9155±0.0013 |
| **M3Care** | 0.8691±0.0011 | 0.4994±0.0061 | 0.2928±0.0437 | 0.7674±0.0513 | 0.1832±0.0369 | 0.9938±0.0027 | 0.9163±0.0014 |
| **MedFuse** | 0.8741±0.0025 | 0.5079±0.0069 | 0.3158±0.0516 | 0.7366±0.0146 | 0.2032±0.0428 | 0.9922±0.0021 | 0.9168±0.0022 |
| **SMIL** | 0.8596±0.0089 | 0.4782±0.0164 | 0.3073±0.0454 | 0.6900±0.0187 | 0.1994±0.0368 | 0.9905±0.0021 | 0.9149±0.0020 |

## C.2 DISEASE BREAKDOWN RESULTS FOR THE PHENOTYPE CLASSIFICATION TASK

We provide the detailed disease breakdown results for the phenotype classification task in Table 27 and Table 28.

## C.3 FAIRNESS BY ATTRIBUTES

We use the AUPRC gap to evaluate the fairness of models across different sensitive attributes on phenotype prediction task (see Table 29). Besides, we plot the performance by subgroups across different sensitive attributes on both the matched subset and the base cohort for phenotype prediction (Fig. 6, Fig. 7); mortality prediction (Fig. 8, Fig. 9); and length-of-stay prediction (Fig. 10, Fig. 11);

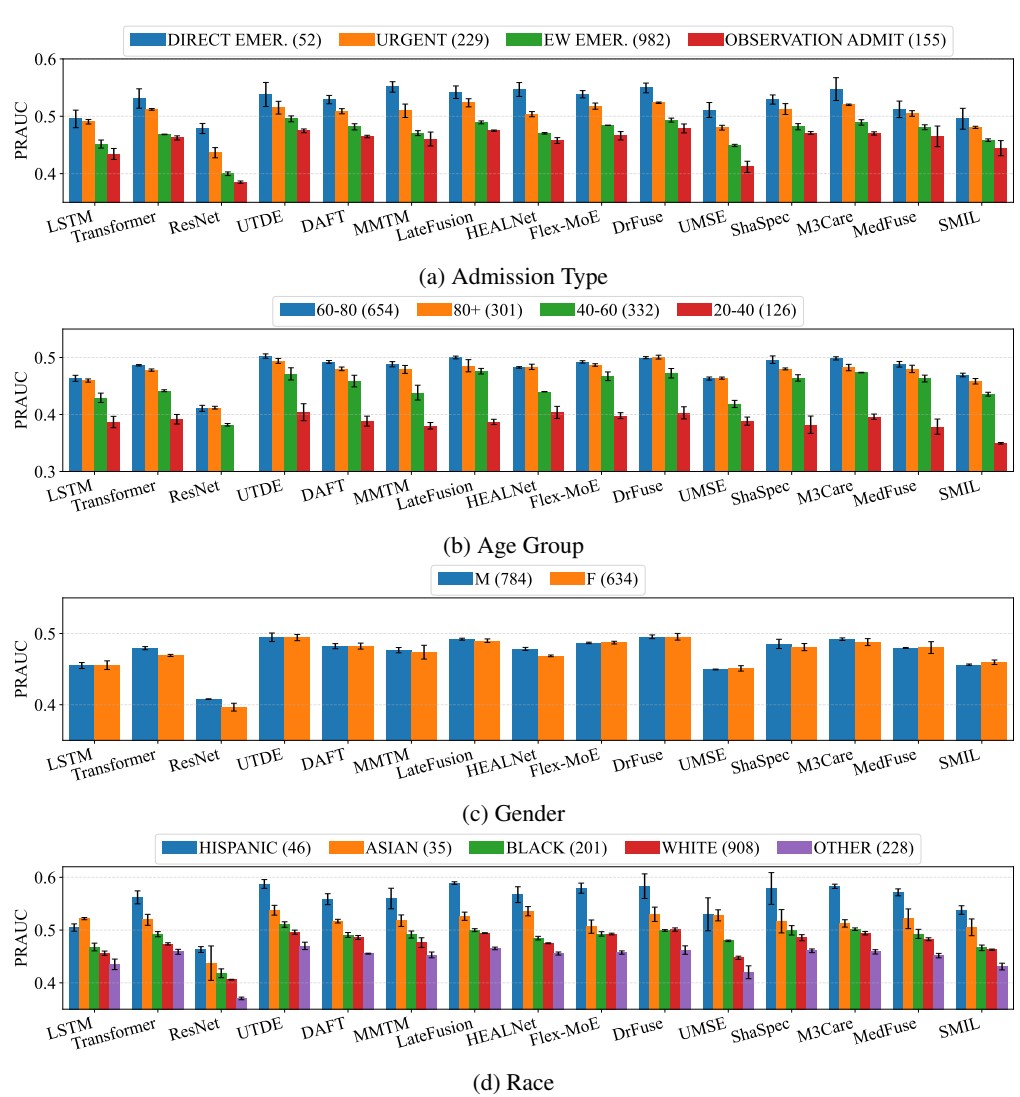

(a) Admission Type

(b) Age Group

(c) Gender

(d) Race

Figure 6: Performance of the phenotype classification on the matched subset grouped by different sensitive attributes.

Table 27: Disease breakdown for the phenotype classification task on the matched subset.

| | LSTM | Transformer | ResNet | UTDE | DAFT | MMTM | LateFusion | HEALNet | Flex-MoE | DrFuse | UMSE | ShaSpec | M3Care | MedFuse | SMIL |
|---|---|---|---|---|---|---|---|---|---|---|---|---|---|---|---|
| **Acute renal failure** | 0.6416±0.0088 | 0.6561±0.0068 | 0.5173±0.0166 | 0.6580±0.0109 | 0.6412±0.0091 | 0.6456±0.0078 | 0.6502±0.0054 | 0.6579±0.0105 | 0.6544±0.0050 | 0.6476±0.0133 | 0.6741±0.0103 | 0.6444±0.0084 | 0.6481±0.0091 | 0.6393±0.0135 | 0.5915±0.0092 |
| **Acute cerebrovascular disease** | 0.4802±0.0134 | 0.4863±0.0053 | 0.2484±0.0129 | 0.4852±0.0058 | 0.4697±0.0077 | 0.4806±0.0200 | 0.4914±0.0134 | 0.5107±0.0349 | 0.4792±0.0132 | 0.5141±0.0029 | 0.4576±0.0124 | 0.4607±0.0128 | 0.4759±0.0123 | 0.4742±0.0056 | 0.4118±0.0217 |
| **Acute myocardial infarction** | 0.2159±0.0307 | 0.2342±0.0006 | 0.1690±0.0126 | 0.2523±0.0017 | 0.2351±0.0169 | 0.2524±0.0074 | 0.2358±0.0124 | 0.2449±0.0036 | 0.2460±0.0180 | 0.2716±0.0112 | 0.2611±0.0099 | 0.2345±0.0229 | 0.2784±0.0337 | 0.2314±0.0067 | 0.1997±0.0126 |
| **Cardiac dysrhythmias** | 0.7323±0.0097 | 0.7439±0.0040 | 0.6056±0.0024 | 0.7371±0.0050 | 0.7322±0.0143 | 0.7378±0.0062 | 0.7458±0.0053 | 0.7299±0.0037 | 0.7349±0.0039 | 0.7312±0.0008 | 0.7264±0.0023 | 0.7264±0.0023 | 0.7345±0.0062 | 0.7219±0.0053 | 0.6716±0.0207 |
| **Chronic kidney disease** | 0.5702±0.0138 | 0.6014±0.0095 | 0.4269±0.0107 | 0.6087±0.0098 | 0.6016±0.0053 | 0.5442±0.0456 | 0.5984±0.0095 | 0.6084±0.0109 | 0.6119±0.0108 | 0.6160±0.0133 | 0.6067±0.0106 | 0.6129±0.0056 | 0.6060±0.0040 | 0.5946±0.0067 | 0.5610±0.0226 |
| **COPD and bronchiectasis** | 0.2844±0.0209 | 0.3102±0.0058 | 0.3780±0.0129 | 0.3846±0.0247 | 0.3654±0.0098 | 0.3805±0.0119 | 0.4021±0.0155 | 0.3414±0.0008 | 0.3720±0.0283 | 0.3983±0.0234 | 0.3520±0.0189 | 0.3760±0.0247 | 0.3931±0.0094 | 0.3799±0.0023 | 0.3594±0.0158 |
| **Surgical Complications** | 0.3752±0.0053 | 0.3766±0.0105 | 0.3446±0.0171 | 0.3778±0.0066 | 0.3678±0.0083 | 0.3732±0.0095 | 0.3786±0.0075 | 0.3691±0.0064 | 0.3742±0.0061 | 0.3806±0.0072 | 0.3798±0.0055 | 0.3868±0.0094 | 0.3737±0.0067 | 0.3735±0.0110 | 0.3542±0.0158 |
| **Conduction disorders** | 0.3954±0.0248 | 0.5079±0.0110 | 0.6478±0.0113 | 0.6824±0.0079 | 0.6863±0.0046 | 0.5935±0.0265 | 0.6874±0.0029 | 0.4614±0.0114 | 0.6849±0.0128 | 0.6732±0.0112 | 0.2357±0.0089 | 0.6806±0.0257 | 0.6786±0.0034 | 0.6832±0.0138 | 0.6575±0.0100 |
| **CHF; nonhypertensive** | 0.6910±0.0107 | 0.7046±0.0032 | 0.6957±0.0027 | 0.7359±0.0048 | 0.7279±0.0016 | 0.7044±0.0152 | 0.7423±0.0055 | 0.6992±0.0025 | 0.7345±0.0025 | 0.7349±0.0111 | 0.6202±0.0176 | 0.7372±0.0030 | 0.7345±0.0070 | 0.7372±0.0106 | 0.7322±0.0092 |
| **CAD** | 0.4910±0.0138 | 0.4867±0.0075 | 0.5152±0.0116 | 0.5497±0.0119 | 0.5444±0.0126 | 0.5430±0.0126 | 0.5365±0.0054 | 0.4967±0.0051 | 0.5505±0.0122 | 0.5533±0.0116 | 0.4849±0.0015 | 0.5479±0.0109 | 0.5500±0.0092 | 0.5441±0.0128 | 0.5194±0.0100 |
| **DM with complications** | 0.5374±0.0223 | 0.6006±0.0057 | 0.2151±0.0066 | 0.5895±0.0029 | 0.5639±0.0226 | 0.5796±0.0258 | 0.5738±0.0096 | 0.6001±0.0048 | 0.5657±0.0149 | 0.5894±0.0221 | 0.6116±0.0165 | 0.5514±0.0115 | 0.5781±0.0089 | 0.5252±0.0336 | 0.4766±0.0080 |
| **DM without complication** | 0.3498±0.0050 | 0.3883±0.0102 | 0.2688±0.0091 | 0.3824±0.0120 | 0.3558±0.0205 | 0.3836±0.0121 | 0.3661±0.0107 | 0.3792±0.0042 | 0.3674±0.0152 | 0.3733±0.0279 | 0.3823±0.0058 | 0.3495±0.0113 | 0.3727±0.0088 | 0.3338±0.0093 | 0.2939±0.0046 |
| **Disorders of lipid metabolism** | 0.5247±0.0054 | 0.5242±0.0037 | 0.5120±0.0089 | 0.5478±0.0032 | 0.5312±0.0090 | 0.5404±0.0190 | 0.5353±0.0122 | 0.5358±0.0093 | 0.5286±0.0023 | 0.5456±0.0021 | 0.5303±0.0024 | 0.5400±0.0168 | 0.5289±0.0035 | 0.5359±0.0068 | 0.5303±0.0096 |
| **Essential hypertension** | 0.5344±0.0109 | 0.5428±0.0037 | 0.5099±0.0115 | 0.5772±0.0108 | 0.5639±0.0140 | 0.5373±0.0072 | 0.5608±0.0089 | 0.5437±0.0158 | 0.5622±0.0177 | 0.5726±0.0148 | 0.5521±0.0051 | 0.5715±0.0166 | 0.5678±0.0139 | 0.5582±0.0126 | 0.5659±0.0049 |
| **Fluid and electrolyte disorders** | 0.6861±0.0058 | 0.6915±0.0052 | 0.6674±0.0057 | 0.6910±0.0065 | 0.6974±0.0066 | 0.6885±0.0046 | 0.7022±0.0026 | 0.6976±0.0031 | 0.6941±0.0012 | 0.6893±0.0029 | 0.6936±0.0053 | 0.6904±0.0103 | 0.7014±0.0029 | 0.6952±0.0059 | 0.6881±0.0072 |
| **Gastrointestinal hemorrhage** | 0.1630±0.0091 | 0.1707±0.0062 | 0.1386±0.0095 | 0.1819±0.0170 | 0.1654±0.0054 | 0.1697±0.0076 | 0.1819±0.0091 | 0.1643±0.0095 | 0.1679±0.0106 | 0.1820±0.0052 | 0.1427±0.0103 | 0.1679±0.0124 | 0.1759±0.0199 | 0.1828±0.0207 | 0.1526±0.0104 |
| **Secondary hypertension** | 0.5289±0.0210 | 0.5667±0.0061 | 0.3947±0.0074 | 0.5665±0.0193 | 0.5581±0.0071 | 0.5052±0.0526 | 0.5570±0.0154 | 0.5793±0.0099 | 0.5725±0.0112 | 0.5776±0.0227 | 0.5748±0.0133 | 0.5839±0.0099 | 0.5654±0.0054 | 0.5518±0.0021 | 0.5217±0.0231 |
| **Other liver diseases** | 0.3288±0.0072 | 0.3246±0.0063 | 0.3338±0.0056 | 0.3824±0.0086 | 0.3670±0.0093 | 0.3409±0.0121 | 0.3693±0.0061 | 0.3309±0.0176 | 0.3519±0.0053 | 0.3548±0.0181 | 0.3003±0.0208 | 0.3648±0.0106 | 0.3628±0.0043 | 0.3635±0.0120 | 0.3591±0.0258 |
| **Other lower respiratory disease** | 0.2194±0.0190 | 0.2055±0.0109 | 0.1979±0.0019 | 0.2085±0.0097 | 0.2058±0.0042 | 0.2119±0.0034 | 0.2101±0.0034 | 0.2088±0.0055 | 0.1941±0.0125 | 0.2047±0.0035 | 0.2102±0.0052 | 0.2171±0.0047 | 0.2102±0.0052 | 0.2173±0.0078 | 0.2036±0.0093 |
| **Other upper respiratory disease** | 0.1670±0.0233 | 0.2196±0.0050 | 0.1638±0.0472 | 0.2036±0.0099 | 0.1849±0.0029 | 0.1933±0.0134 | 0.2081±0.0194 | 0.2068±0.0141 | 0.1833±0.0111 | 0.1940±0.0090 | 0.1551±0.0170 | 0.1716±0.0161 | 0.1835±0.0155 | 0.1764±0.0212 | 0.1289±0.0095 |
| **Pleurisy; pneumothorax** | 0.1599±0.0016 | 0.1769±0.0064 | 0.2083±0.0146 | 0.2022±0.0108 | 0.1970±0.0166 | 0.1778±0.0204 | 0.2035±0.0080 | 0.1706±0.0066 | 0.1891±0.0095 | 0.2004±0.0145 | 0.1687±0.0140 | 0.1743±0.0078 | 0.1958±0.0116 | 0.1945±0.0294 | 0.1929±0.0035 |
| **Pneumonia** | 0.4551±0.0059 | 0.4701±0.0134 | 0.4116±0.0094 | 0.4949±0.0162 | 0.4796±0.0172 | 0.4649±0.0054 | 0.4929±0.0032 | 0.4775±0.0097 | 0.4868±0.0135 | 0.5000±0.0237 | 0.4463±0.0109 | 0.4709±0.0074 | 0.4933±0.0071 | 0.4754±0.0010 | 0.4671±0.0188 |
| **Respiratory failure** | 0.6482±0.0095 | 0.6642±0.0088 | 0.5871±0.0106 | 0.6626±0.0071 | 0.6584±0.0104 | 0.6498±0.0010 | 0.6649±0.0093 | 0.6562±0.0023 | 0.6629±0.0081 | 0.6628±0.0103 | 0.6480±0.0063 | 0.6604±0.0169 | 0.6616±0.0065 | 0.6533±0.0044 | 0.6550±0.0069 |
| **Septicemia (except in labor)** | 0.5531±0.0033 | 0.5658±0.0077 | 0.4167±0.0045 | 0.5698±0.0055 | 0.5366±0.0135 | 0.5496±0.0118 | 0.5560±0.0057 | 0.5437±0.0048 | 0.5694±0.0025 | 0.5756±0.0145 | 0.5459±0.0057 | 0.5556±0.0172 | 0.5669±0.0073 | 0.5443±0.0145 | 0.5353±0.0073 |
| **Shock** | 0.5662±0.0089 | 0.5753±0.0050 | 0.4182±0.0216 | 0.5682±0.0137 | 0.5601±0.0048 | 0.5608±0.0098 | 0.5745±0.0144 | 0.5717±0.0101 | 0.5759±0.0051 | 0.5777±0.0046 | 0.5637±0.0077 | 0.5548±0.0086 | 0.5653±0.0118 | 0.5551±0.0082 | 0.5606±0.0145 |
| **Average Rank** | 11.32 | 7.68 | 13.36 | 4.12 | 8.36 | 8.48 | 4.72 | 7.96 | 6.64 | 4.24 | 9.72 | 7.36 | 5.96 | 8.48 | 11.6 |

Table 28: Disease breakdown for the phenotype classification task on the base cohort.

| | LSTM | Transformer | ResNet | UTDE | DAFT | MMTM | LateFusion | HEALNet | Flex-MoE | DrFuse | UMSE | ShaSpec | M3Care | MedFuse | SMIL |
|---|---|---|---|---|---|---|---|---|---|---|---|---|---|---|---|
| **Acute renal failure** | 0.6612±0.0051 | 0.6720±0.0009 | 0.4235±0.0045 | 0.6733±0.0037 | 0.6596±0.0048 | 0.6643±0.0059 | 0.6701±0.0054 | 0.6679±0.0054 | 0.6696±0.0075 | 0.6721±0.0084 | 0.6750±0.0021 | 0.6686±0.0015 | 0.6676±0.0036 | 0.6720±0.0069 | 0.6412±0.0065 |
| **Acute cerebrovascular disease** | 0.4320±0.0141 | 0.4351±0.0067 | 0.1367±0.0161 | 0.4571±0.0085 | 0.4473±0.0099 | 0.4489±0.0141 | 0.4496±0.0015 | 0.4558±0.0124 | 0.4378±0.0051 | 0.4482±0.0123 | 0.4308±0.0131 | 0.4507±0.0111 | 0.4591±0.0074 | 0.4389±0.0069 | 0.4001±0.0069 |
| **Acute myocardial infarction** | 0.2744±0.0201 | 0.2950±0.0041 | 0.1270±0.0018 | 0.3166±0.0187 | 0.2761±0.0291 | 0.3022±0.0146 | 0.3013±0.0139 | 0.2941±0.0071 | 0.3041±0.0033 | 0.2915±0.0042 | 0.2825±0.0147 | 0.3112±0.0094 | 0.3190±0.0045 | 0.2769±0.0217 | 0.2312±0.0109 |
| **Cardiac dysrhythmias** | 0.7135±0.0029 | 0.7144±0.0029 | 0.4658±0.0039 | 0.7137±0.0013 | 0.7069±0.0061 | 0.7033±0.0015 | 0.7140±0.0057 | 0.7104±0.0044 | 0.7158±0.0018 | 0.7193±0.0064 | 0.7589±0.0106 | 0.7155±0.0025 | 0.7172±0.0025 | 0.6960±0.0030 | 0.6658±0.0053 |
| **Chronic kidney disease** | 0.5957±0.0015 | 0.6060±0.0028 | 0.3046±0.0086 | 0.5991±0.0007 | 0.6067±0.0067 | 0.5762±0.0130 | 0.6067±0.0069 | 0.6045±0.0022 | 0.6063±0.0092 | 0.6050±0.0037 | 0.6058±0.0022 | 0.6008±0.0084 | 0.6067±0.0057 | 0.5949±0.0032 | 0.5693±0.0089 |
| **COPD and bronchiectasis** | 0.3394±0.0095 | 0.3617±0.0058 | 0.2265±0.0029 | 0.3856±0.0021 | 0.3712±0.0031 | 0.3409±0.0062 | 0.3782±0.0043 | 0.3659±0.0080 | 0.3907±0.0018 | 0.3873±0.0068 | 0.3734±0.0052 | 0.3753±0.0039 | 0.3868±0.0051 | 0.3506±0.0057 | 0.3268±0.0081 |
| **Surgical Complications** | 0.3955±0.0071 | 0.3940±0.0037 | 0.2606±0.0012 | 0.3995±0.0085 | 0.3861±0.0059 | 0.4028±0.0039 | 0.3916±0.0040 | 0.4011±0.0029 | 0.3876±0.0042 | 0.3956±0.0105 | 0.4055±0.0020 | 0.3942±0.0023 | 0.3988±0.0012 | 0.3996±0.0068 | 0.3779±0.0070 |
| **Conduction disorders** | 0.4352±0.0134 | 0.4791±0.0068 | 0.2650±0.0053 | 0.5276±0.0138 | 0.5197±0.0078 | 0.4375±0.0030 | 0.5412±0.0083 | 0.4597±0.0037 | 0.5414±0.0009 | 0.5291±0.0069 | 0.2570±0.0073 | 0.5291±0.0181 | 0.5370±0.0061 | 0.4285±0.0103 | 0.4378±0.0226 |
| **CHF; nonhypertensive** | 0.6719±0.0109 | 0.6861±0.0036 | 0.4454±0.0033 | 0.6896±0.0048 | 0.6801±0.0067 | 0.6728±0.0006 | 0.6990±0.0040 | 0.6786±0.0082 | 0.6837±0.0080 | 0.6944±0.0047 | 0.6123±0.0006 | 0.6842±0.0045 | 0.6890±0.0007 | 0.6748±0.0044 | 0.6738±0.0040 |
| **CAD** | 0.4924±0.0015 | 0.5041±0.0038 | 0.3424±0.0052 | 0.5181±0.0082 | 0.5212±0.0091 | 0.5030±0.0086 | 0.5137±0.0024 | 0.5074±0.0039 | 0.5196±0.0023 | 0.5217±0.0026 | 0.5009±0.0068 | 0.5198±0.0004 | 0.5194±0.0015 | 0.4985±0.0087 | 0.4978±0.0020 |
| **DM with complications** | 0.5718±0.0062 | 0.5709±0.0051 | 0.1558±0.0058 | 0.5844±0.0034 | 0.5751±0.0145 | 0.5699±0.0041 | 0.5743±0.0080 | 0.5740±0.0121 | 0.5878±0.0067 | 0.5814±0.0033 | 0.5824±0.0018 | 0.5885±0.0075 | 0.5896±0.0057 | 0.5659±0.0049 | 0.5416±0.0083 |
| **DM without complication** | 0.3758±0.0075 | 0.3750±0.0030 | 0.1993±0.0017 | 0.3839±0.0023 | 0.3633±0.0037 | 0.3700±0.0070 | 0.3808±0.0044 | 0.3632±0.0033 | 0.3676±0.0068 | 0.3692±0.0035 | 0.3816±0.0111 | 0.3787±0.0023 | 0.3866±0.0045 | 0.3631±0.0029 | 0.3289±0.0055 |
| **Disorders of lipid metabolism** | 0.5626±0.0030 | 0.5614±0.0041 | 0.4296±0.0039 | 0.5600±0.0010 | 0.5571±0.0075 | 0.5628±0.0035 | 0.5653±0.0028 | 0.5559±0.0005 | 0.5543±0.0019 | 0.5602±0.0032 | 0.5570±0.0012 | 0.5600±0.0030 | 0.5574±0.0012 | 0.5608±0.0028 | 0.5620±0.0030 |
| **Essential hypertension** | 0.5628±0.0006 | 0.5717±0.0051 | 0.4261±0.0075 | 0.5769±0.0015 | 0.5724±0.0044 | 0.5605±0.0109 | 0.5688±0.0048 | 0.5643±0.0043 | 0.5728±0.0056 | 0.5746±0.0043 | 0.5735±0.0062 | 0.5749±0.0039 | 0.5835±0.0046 | 0.5668±0.0121 | 0.5595±0.0023 |
| **Fluid and electrolyte disorders** | 0.6879±0.0006 | 0.6904±0.0022 | 0.5400±0.0065 | 0.6900±0.0034 | 0.6806±0.0022 | 0.6831±0.0035 | 0.6913±0.0023 | 0.6894±0.0017 | 0.6883±0.0012 | 0.6909±0.0061 | 0.6856±0.0004 | 0.6881±0.0056 | 0.6899±0.0003 | 0.6889±0.0028 | 0.6751±0.0013 |
| **Gastrointestinal hemorrhage** | 0.1962±0.0152 | 0.2103±0.0071 | 0.0855±0.0000 | 0.2083±0.0076 | 0.1875±0.0136 | 0.2022±0.0150 | 0.1996±0.0014 | 0.1961±0.0097 | 0.2051±0.0049 | 0.1921±0.0084 | 0.1842±0.0058 | 0.2112±0.0027 | 0.2198±0.0033 | 0.1807±0.0164 | 0.1465±0.0016 |
| **Secondary hypertension** | 0.5662±0.0072 | 0.5717±0.0009 | 0.3041±0.0071 | 0.5696±0.0021 | 0.5712±0.0078 | 0.5594±0.0061 | 0.5743±0.0072 | 0.5647±0.0078 | 0.5727±0.0137 | 0.5726±0.0031 | 0.5723±0.0028 | 0.5687±0.0028 | 0.5707±0.0060 | 0.5713±0.0024 | 0.5599±0.0061 |
| **Other liver diseases** | 0.3416±0.0072 | 0.3557±0.0073 | 0.2239±0.0020 | 0.3674±0.0060 | 0.3581±0.0117 | 0.3494±0.0048 | 0.3649±0.0049 | 0.3630±0.0047 | 0.3580±0.0085 | 0.3651±0.0085 | 0.3264±0.0040 | 0.3661±0.0063 | 0.3626±0.0112 | 0.3369±0.0089 | 0.3377±0.0023 |
| **Other lower respiratory disease** | 0.2021±0.0085 | 0.2147±0.0074 | 0.1479±0.0036 | 0.2160±0.0041 | 0.2030±0.0067 | 0.2494±0.0037 | 0.2048±0.0031 | 0.2542±0.0085 | 0.2169±0.0074 | 0.2112±0.0037 | 0.2110±0.0041 | 0.2091±0.0064 | 0.2079±0.0062 | 0.2142±0.0065 | 0.1775±0.0067 |
| **Other upper respiratory disease** | 0.2491±0.0111 | 0.2704±0.0112 | 0.0986±0.0107 | 0.2704±0.0068 | 0.2497±0.0080 | 0.2455±0.0058 | 0.2773±0.0039 | 0.2542±0.0085 | 0.2733±0.0083 | 0.2733±0.0128 | 0.2133±0.0061 | 0.2864±0.0108 | 0.2813±0.0179 | 0.2324±0.0150 | 0.1651±0.0285 |
| **Pleurisy; pneumothorax** | 0.1943±0.0123 | 0.1959±0.0058 | 0.1250±0.0032 | 0.2024±0.0070 | 0.1983±0.0025 | 0.1976±0.0055 | 0.2040±0.0018 | 0.1940±0.0069 | 0.2048±0.0036 | 0.2069±0.0067 | 0.1913±0.0025 | 0.2024±0.0033 | 0.2072±0.0065 | 0.2019±0.0111 | 0.1714±0.0044 |
| **Pneumonia** | 0.4086±0.0052 | 0.4284±0.0048 | 0.2790±0.0103 | 0.4256±0.0020 | 0.4208±0.0094 | 0.4099±0.0024 | 0.4357±0.0014 | 0.4202±0.0078 | 0.4315±0.0107 | 0.4285±0.0046 | 0.4256±0.0030 | 0.4251±0.0066 | 0.4347±0.0051 | 0.4248±0.0006 | 0.4164±0.0050 |
| **Respiratory failure** | 0.6093±0.0079 | 0.6151±0.0076 | 0.3890±0.0015 | 0.6217±0.0061 | 0.6153±0.0083 | 0.6138±0.0055 | 0.6226±0.0033 | 0.6109±0.0048 | 0.6170±0.0028 | 0.6280±0.0021 | 0.6178±0.0037 | 0.6250±0.0058 | 0.6224±0.0037 | 0.5986±0.0011 | 0.5976±0.0096 |
| **Septicemia (except in labor)** | 0.5617±0.0063 | 0.5697±0.0024 | 0.3067±0.0089 | 0.5653±0.0022 | 0.5587±0.0071 | 0.5570±0.0083 | 0.5683±0.0059 | 0.5608±0.0034 | 0.5676±0.0044 | 0.5747±0.0045 | 0.5685±0.0093 | 0.5681±0.0039 | 0.5694±0.0053 | 0.5691±0.0041 | 0.5358±0.0017 |
| **Shock** | 0.6077±0.0011 | 0.6195±0.0046 | 0.2957±0.0033 | 0.6095±0.0050 | 0.6010±0.0159 | 0.6083±0.0038 | 0.6205±0.0024 | 0.6147±0.0033 | 0.6141±0.0084 | 0.6192±0.0039 | 0.6167±0.0059 | 0.6174±0.0009 | 0.6234±0.0023 | 0.6100±0.0061 | 0.6029±0.0048 |
| **Average Rank** | 10.76 | 6.6 | 14.96 | 5.12 | 9.16 | 9.52 | 4.68 | 8.84 | 5.8 | 4.4 | 8.56 | 5.32 | 3.72 | 9.48 | 13.08 |

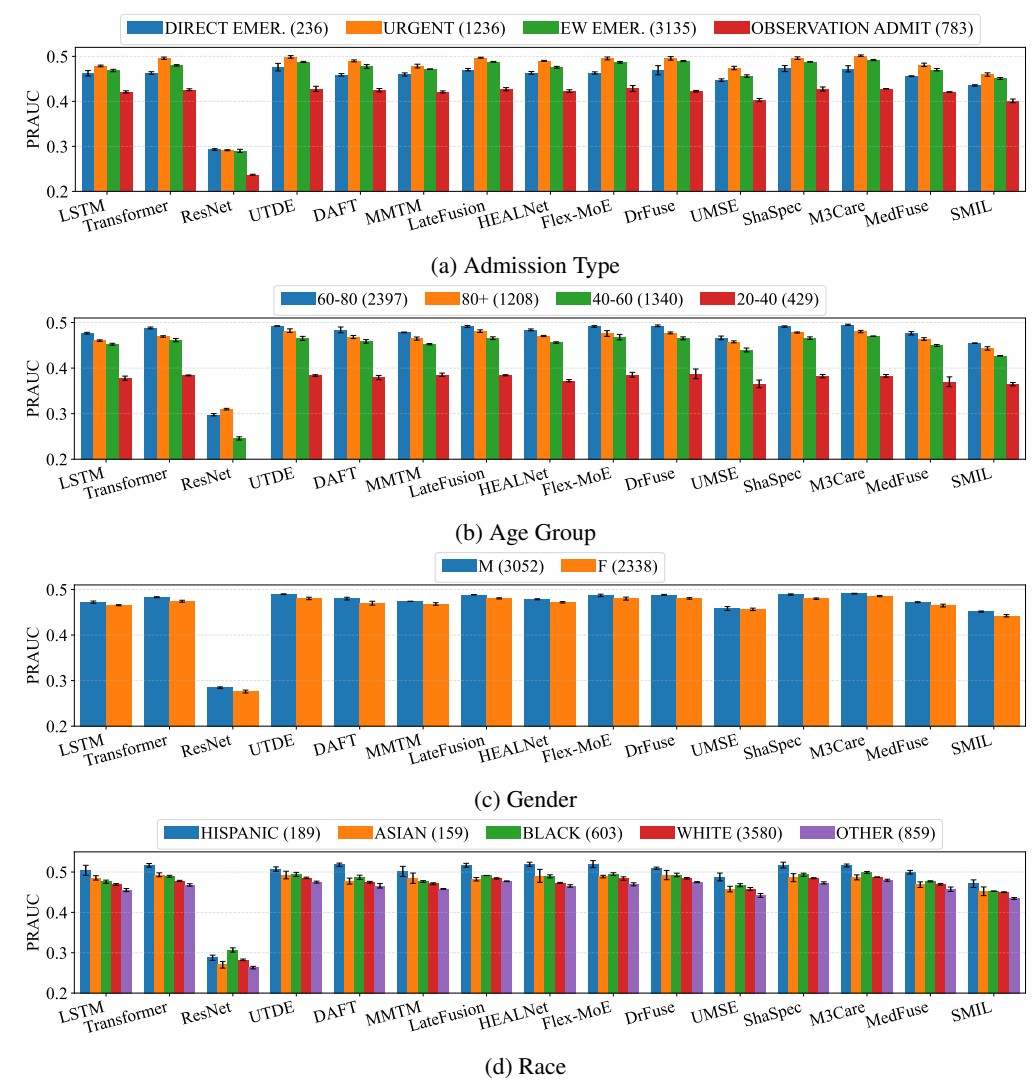

Figure 7: Performance of the phenotype classification on the base cohort grouped by different sensitive attributes.

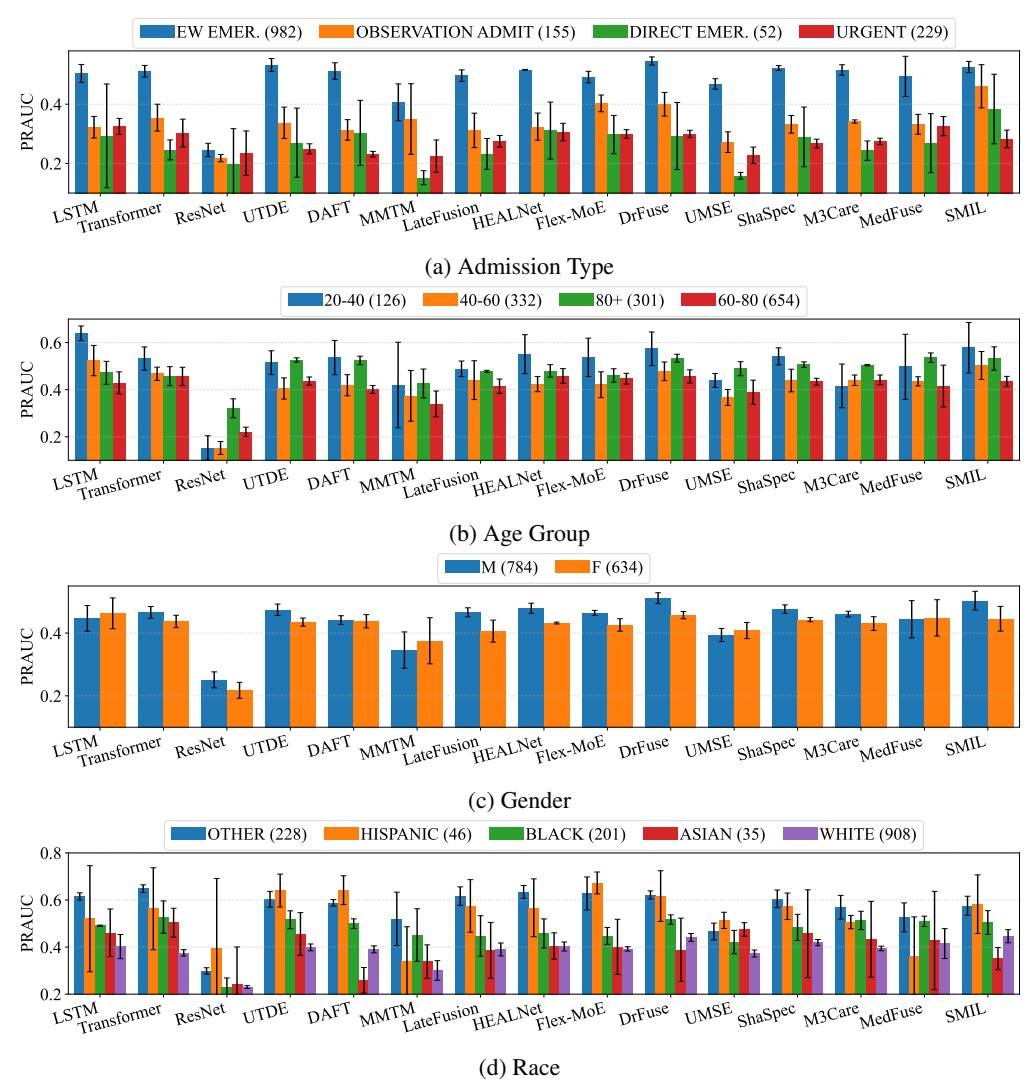

Figure 8: Performance of the mortality prediction on the matched subset grouped by different sensitive attributes.

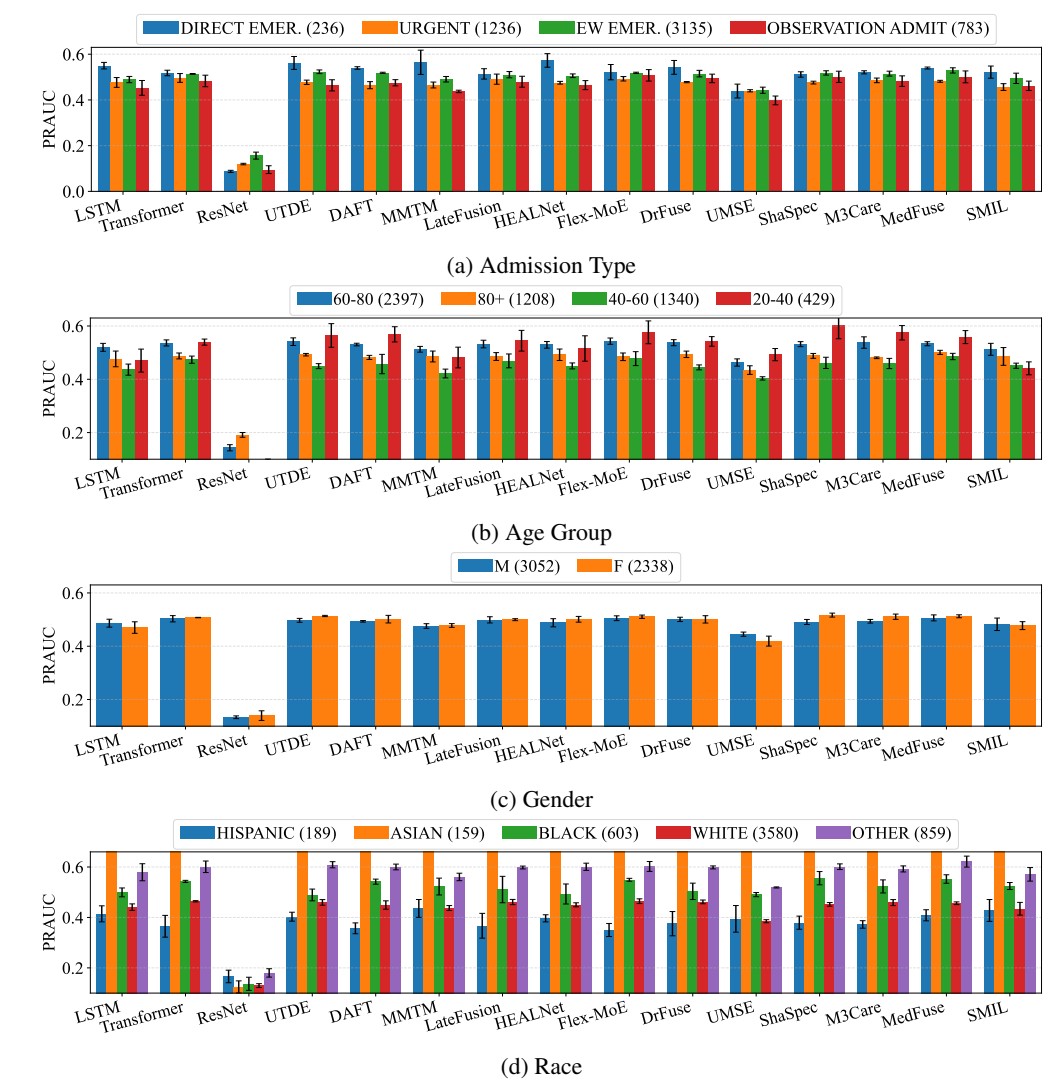

Figure 9: Performance of the mortality prediction on the base cohort grouped by different sensitive attributes.

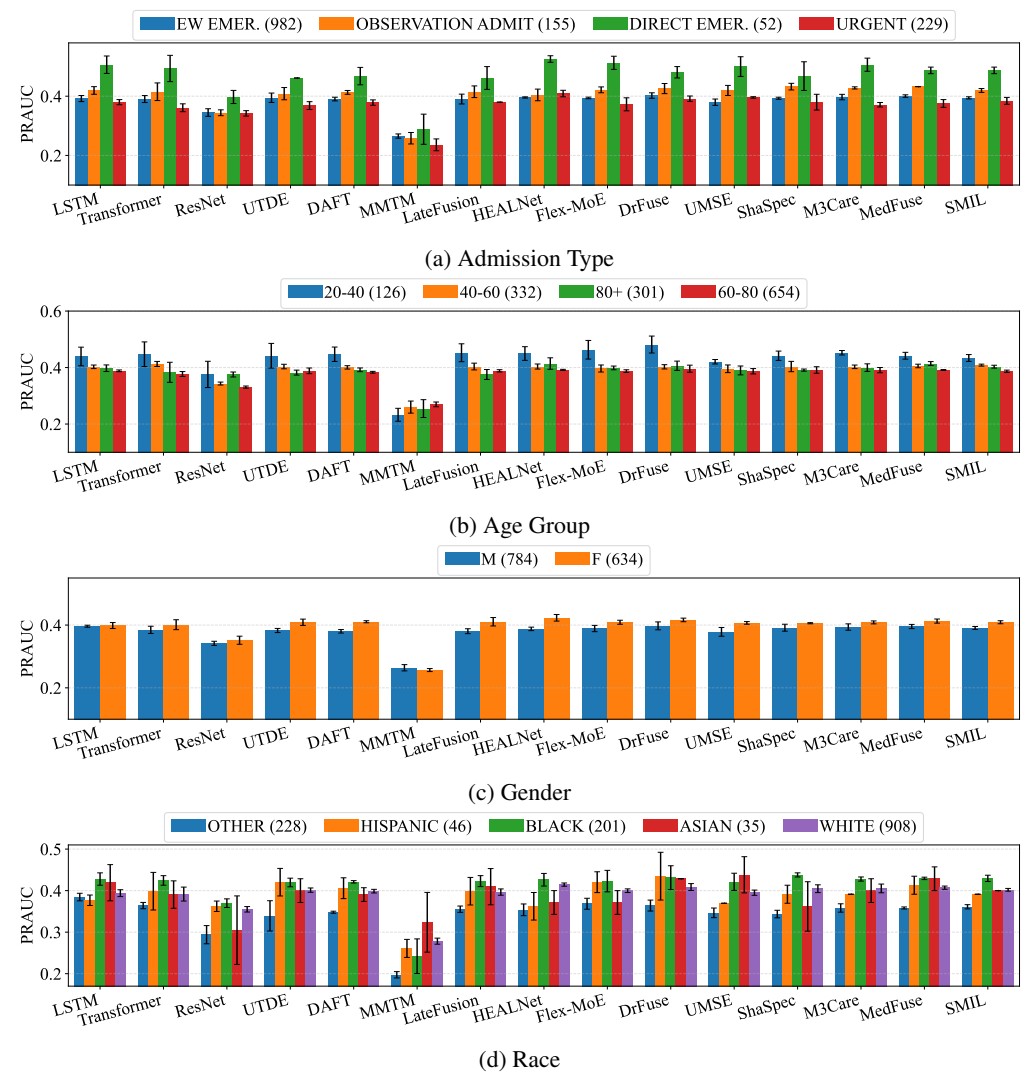

Figure 10: Performance of the length-of-stay prediction on the matched subset grouped by different sensitive attributes.

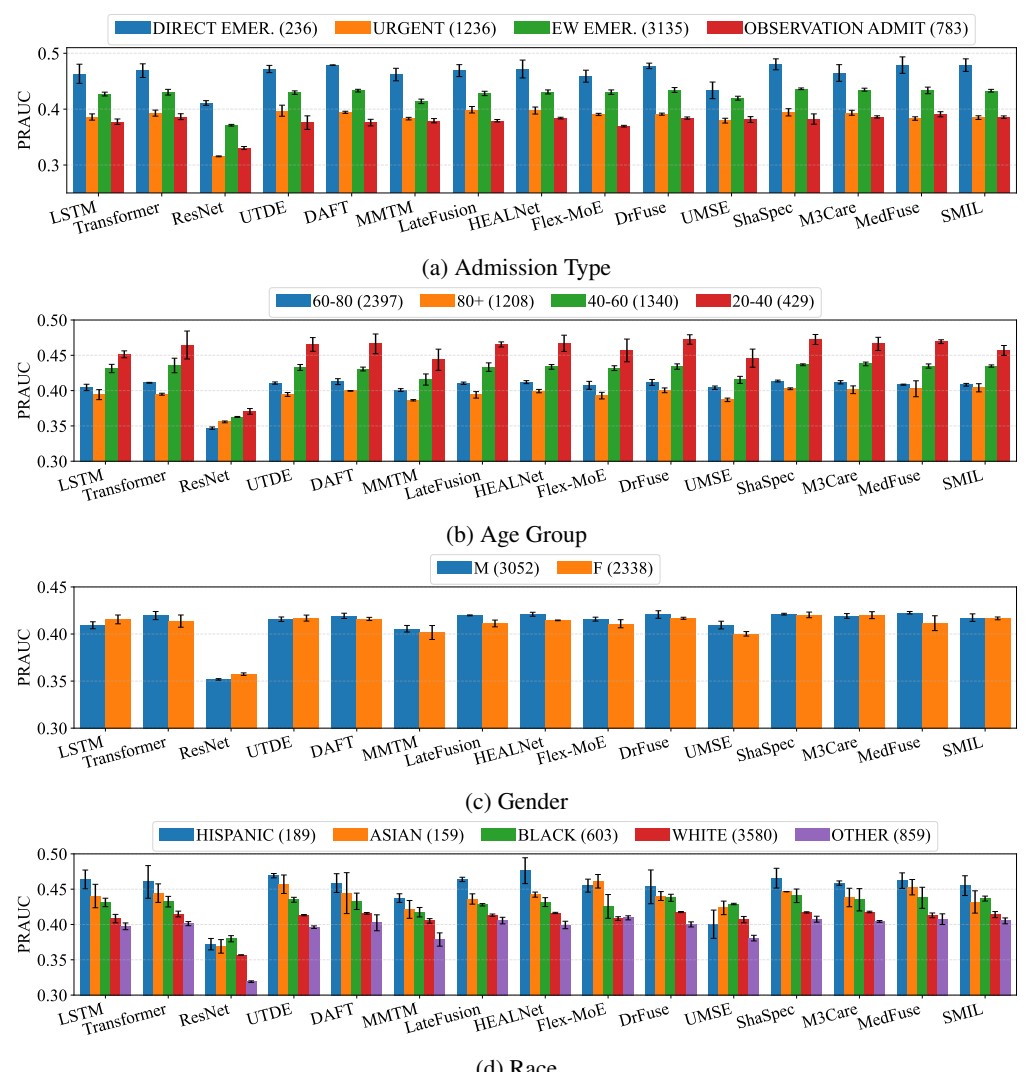

Figure 11: Performance of the length-of-stay prediction on the base cohort grouped by different sensitive attributes.

Table 29: Fairness performance on phenotype prediction task measured as the PRAUC gap (best subgroup – worst subgroup) across different sensitive attributes (the larger gap, the more unfair)

| | Race | | Gender | | Age Group | | Admission Type | |
|---|---|---|---|---|---|---|---|---|
| | Matched | Full | Matched | Full | Matched | Full | Matched | Full |
| LSTM | 0.0867 | 0.0489 | 0.0005 | 0.0067 | 0.0767 | 0.1649 | 0.0611 | 0.0576 |
| Transformer | 0.1032 | 0.0485 | 0.0102 | 0.0089 | 0.0947 | 0.1189 | 0.0687 | 0.0700 |
| ResNet | 0.0927 | 0.0437 | 0.0113 | 0.0084 | 0.1205 | 0.1308 | 0.0936 | 0.0559 |
| UTDE | 0.1174 | 0.0323 | 0.0004 | 0.0094 | 0.0986 | 0.1523 | 0.0634 | 0.0710 |
| DAFT | 0.1032 | 0.0522 | 0.0001 | 0.0101 | 0.1039 | 0.1574 | 0.0644 | 0.0650 |
| MMTM | 0.1069 | 0.0433 | 0.0028 | 0.0058 | 0.1080 | 0.1292 | 0.0908 | 0.0574 |
| LateFusion | 0.1238 | 0.0397 | 0.0021 | 0.0080 | 0.1131 | 0.1070 | 0.0672 | 0.0700 |
| HEALNet | 0.1116 | 0.0536 | 0.0098 | 0.0067 | 0.0799 | 0.1495 | 0.0890 | 0.0672 |
| Flex-MoE | 0.1220 | 0.0501 | 0.0004 | 0.0070 | 0.0943 | 0.1094 | 0.0724 | 0.0668 |
| DrFuse | 0.1211 | 0.0346 | 0.0000 | 0.0080 | 0.0978 | 0.1295 | 0.0706 | 0.0730 |
| UMSE | 0.1099 | 0.0457 | 0.0015 | 0.0019 | 0.0754 | 0.1811 | 0.0992 | 0.0712 |
| ShaSpec | 0.1181 | 0.0443 | 0.0045 | 0.0092 | 0.1141 | 0.1282 | 0.0583 | 0.0685 |
| M3Care | 0.1243 | 0.0365 | 0.0041 | 0.0054 | 0.1024 | 0.1122 | 0.0773 | 0.0735 |
| MedFuse | 0.1067 | 0.0421 | 0.0065 | 0.0074 | 0.1028 | 0.1571 | 0.0662 | 0.0604 |
| SMIL | 0.1072 | 0.0368 | 0.0033 | 0.0092 | 0.1198 | 0.0973 | 0.0513 | 0.0585 |

