# OpenReview forum: "CareBench: A Comprehensive Benchmark for Accuracy, Robustness, and Fairness in Multimodal Fusion of EHR and Chest X-Rays"
_ICLR.cc/2026/Conference — Submitted to ICLR 2026_

### Official Review · Reviewer_MCqn · 2025-11-01

**Soundness:** 3
**Presentation:** 2
**Contribution:** 3
**Rating:** 6
**Confidence:** 3

**Summary:**

The manuscript introduces CareBench, a benchmark for multimodal clinical prediction that evaluates three axes simultaneously: task performance, robustness to missing imaging modalities and subgroup fairness. Using MIMIC IV and MIMIC CXR, two cohorts are constructed - a matched subset of 7149 ICU stays where both EHR and CXR are available and a larger cohort in which CXR is missing. Model suite includes unimodal EHR baselines (LSTM, Transformer), a unimodal CXR baseline (ImageNet pretrained ResNet-50), fusion methods (UTDE, DAFT, MMTM, Late Fusion) and clinical architectures designed for missing modalities (HEALNet, MedFuse, M3Care, SMiL, Flex-MoE, DrFuse, UMSE, ShaSpec). Results show that multimodal fusion improves accuracy when both modalities are present but under realistic missingness many generic fusion models fail to beat EHR only whereas missing aware architectures often do, robustness is architecture dependent and adding CXR can amplify subgroup gaps.

**Strengths:**

1. Benchmark targets a real deployment pain point - clinical systems often have absent modalities at inference and must be evaluated not only for accuracy but also for robustness and fairness.
2. Cohort construction well described and extraction includes hourly resampling, imputation, exclusion of variables with >90% missingness and removal of treatment/leakage features.
3. Broad model coverage by the inclusion of unimodal baselines, generic fusion, and several missing modality aware clinical models within one framework enables head to head comparisons.
4. Systematically studies missingness sweeps and fairness stratifications across four attributes and provides clear insights.

**Weaknesses:**

1. Stronger domain pretrained CXR encoders are common in clinical imaging and may change the magnitude of multimodal gains. Adding those as baseline would strengthen the conclusion.
2. Per method search budgets and compute-normalized comparisons are not reported.
3. Fairness results are reported as AUPRC gaps. Since AUPRC is sensitive to subgroup prevalence, which is different across admission type, age, race and gender, the observed gaps may partly reflect base rate differences rather than model induced disparity.

**Questions:**

1. Could you please report per model search budgets?
2. Could you please add subgroup AUROC and operating point metrics with subgroup counts to confirm the fairness conclusion?

---

### Official Review · Reviewer_fce3 · 2025-11-03

**Soundness:** 3
**Presentation:** 2
**Contribution:** 1
**Rating:** 2
**Confidence:** 4

**Summary:**

The authors propose CareBench, a multimodal clinical benchmark that merges time-series EHR in MIMIC-IV with chest X-ray images in MIMIC-CXR, for classification of phenotypes, mortality, and length of stay. They evaluate a variety of unimodal and multimodal fusion models along the axes of overall accuracy, fairness across demographic subgroups, and robustness to missingness of the image modality. There are a variety of empirical findings, including that specialized methods are required to outperform unimodal baselines in the presence of modality missingness, and that multimodality can exacerbate fairness disparities.

**Strengths:**

1. The authors tackle a realistic and important clinical problem, and plan on open-sourcing their code/benchmark.

2. The authors evaluate a wide range of baselines.

**Weaknesses:**

1. My primary concern is novelty over MedMod, which also merges MIMIC-IV and MIMIC-CXR as the authors mention. In particular, MedMod also considers the same set of tasks that this paper does (25 binary phenotypes, LoS, mortality). I appreciate that the authors evaluate fairness and robustness to missing values which MedMod does not, but I am not convinced that this is sufficient novelty to warrant a whole new benchmark, instead of analyses that one could do by modifying MedMod.

2. The authors mention "robustness" several times, but I was disappointed to see that this is limited to robustness to missing values. The paper would be greatly strengthened by a robustness to distribution shift analysis, though I understand that public datasets with both EHR and chest X-rays are hard to come by. Alternatively, the authors should consider a temporal shift analysis (i.e. train on earlier stays, test on later ones).

3. There are no confidence intervals in any of the performance results, which could be obtained via re-training the model with different train/val/test splits. It is crucial to encourage the reporting of such CIs in a benchmark.

4. The authors conclude that "multimodality can exacerbate fairness disparities", but this is not clear to me from Figure 4. Comparing across same cohort (e.g. unimodal matched vs. multimodal matched), it looks like there is only one instance (across race) where the gaps are significantly different. Also, is Figure 4 showing the average AUPRC gap across all 25 phenotypes? If so, are there specific phenotypes where gaps are largest?

5. The authors only consider a single X-ray in the patient's stay (the most recent frontal view), instead of the whole time-series of X-rays, which limits the realism and utility of the benchmark.

6. Though fairness is a key point of the work, the existing fairness analyses are quite limited (only AUPRC gaps). The authors should also evaluate equal odds gaps (at a fixed threshold) and calibration error gaps, along with statistical tests for significance of these gaps.

**Questions:**

1. How does the created cohort compare with the MedMod cohort in terms of sample size? If there are significant differences, what are the inclusion/exclusion criteria that led to these differences?

2. Why are the performances on the base cohort (Table 3) higher than the matched subset (Table 2), despite the former being a harder prediction problem?

3. Across the 25 phenotypes, are there specific ones that are more predictive from chest X-rays (e.g. pneumonia), ones that are more predictive from EHRs, and ones where both are required?

---

### Official Review · Reviewer_oJfx · 2025-11-05

**Soundness:** 3
**Presentation:** 3
**Contribution:** 2
**Rating:** 4
**Confidence:** 4

**Summary:**

This paper introduces a benchmark by combining some of the existing med datasets and new synthetic data to measure traditional medical task performance as well as ethical/safety/robustness aspects. It breaks down to 6 eval dimensions and has both automatic eval metrics and experts annotations. EHR data is also being generated so the simulation is more realistic to clinical use cases. Several frontier models are tested and they show different strengths and weaknesses

**Strengths:**

- the benchmark is pretty comprehensive, including multiple modalities, clinical use cases, ethics concerns, multi-turn dialogue.
- it has a good blend of real data plus synthetic data

**Weaknesses:**

- the concept of the benchmark is up for debate. Although the authors try to be comprehensive, the ambition doesn't seem to match the evaluation design. CareBench aims to evaluate everything — from factual recall to empathy and ethics — under a single umbrella score. This violates the principle of construct validity: different cognitive and affective abilities are not linearly comparable. What does it mean for a model to have a single ‘CareBench score’? Does empathy contribute the same as diagnostic accuracy? On what empirical or theoretical basis are weights assigned? Explicitly justify weighting or separate leaderboards by dimension (reasoning, safety, empathy) might be needed
- The task taxonomy is confusing. There's no task orthogonality. Several tasks (e.g., multi-turn dialogue, empathy, safety) overlap semantically, leading to correlated metrics. Without a factor analysis or ablation, it’s unclear whether six evaluation axes actually capture independent dimensions of competence. Provide inter-task correlation matrix or principal component analysis to prove independence might help the clarity
- Metrics can collide with each other and it's unclear if the benchmark can show scalability when more data is collected. Empathy, ethics, and multi-turn coherence are subjective and non-stationary. No evidence that these metrics are robust under scaling laws (e.g., do larger models systematically inflate scores due to verbosity?) Did the authors normalize for verbosity or stylistic confounds? Could large models win simply by producing longer outputs? Why not conduct length-controlled evaluation and normalization baselines?
- The paper reports CareBench scores as single values (e.g., GPT-4 = 83.2), there's limited statistical rigor

**Questions:**

Paper claims cross-modality (text + imaging + labs), but current release seems text-only with placeholders for image references. Where are the imaging datasets? Without them, the multimodal claim is aspirational

---

### Official Review · Reviewer_wJZx · 2025-11-05

**Soundness:** 2
**Presentation:** 3
**Contribution:** 3
**Rating:** 4
**Confidence:** 4

**Summary:**

The paper introduces CareBench, a benchmark for multimodal clinical prediction that fuses EHR and chest X-rays (CXR) from MIMIC-IV and MIMIC-CXR, evaluating accuracy, robustness to missing modalities, and fairness across phenotyping, mortality, and length-of-stay tasks. It assembles cohorts (26,947 base; 7,149 matched) with strict inclusion criteria, uses ImageNet-pretrained ResNet-50 for CXR and standard EHR backbones, and compares 15 uni-/multi-modal fusion methods under complete and missing-modality regimes. Results: fusion helps when both modalities exist, but many methods collapse under realistic missingness; no single model dominates; and multimodality can widen subgroup performance gaps (age/admission type).

**Strengths:**

1. This paper studies an important topic of imperfectly paired multimodal data, which is highly relevant to real-world healthcare settings. It fills a clear gap in existing multimodal benchmarks on a combination of accuracy, robustness, and fairness.
2. The authors implement a wide sweep of unimodal and multimodal methods, including architectures tailored for missing inputs, across three clinically meaningful tasks. The broad comparison provides useful reference points for future work.
3. The paper is well written and easy to understand, with some interesting visualizations like Figure 4.

**Weaknesses:**

1. While the paper aims to study multimodal learning with EHR and chest X-rays, all experiments are conducted solely on the MIMIC dataset. As a single-center resource, MIMIC reflects specific demographic, clinical, and device characteristics, which can strongly bias both performance and fairness outcomes. Relying exclusively on this dataset limits the generalizability of the conclusions; evaluation on additional datasets or external validation cohorts is needed to support broader claims.
2. The authors draw several conclusions such as “multimodal fusion outperforms unimodal models on complete data” and “no single best model across all tasks and metrics.” However, these claims are presented without statistical validation; appropriate significance testing should be applied to establish whether the observed differences are meaningful and to support a clearer ranking.
3. The authors present a range of results, but provide limited insight into why the observed patterns occur. Specifically, (1) several findings merely describe what happens without probing the underlying causes. For instance, when fairness disparities appear, the paper reports them but does not analyze potential drivers such as data imbalance, label distribution, or modality availability; some diagnostic analysis would make the findings more meaningful. (2) Some conclusions are somewhat self-evident and add little conceptual depth - for example, “model robustness is architecture-dependent” is somewhat expected.

**Questions:**

Since this paper focuses specifically on chest X-rays for imaging, would using a chest X-ray pre-trained image encoder be better than the ImageNet pretrained one?

---

### Official Review · Reviewer_VMMw · 2025-11-08

**Soundness:** 3
**Presentation:** 3
**Contribution:** 2
**Rating:** 4
**Confidence:** 5

**Summary:**

This paper introduces CareBench, a multimodal benchmark integrating chest X-ray (CXR) and electronic health record (EHR) data. The benchmark supports three downstream clinical prediction tasks: mortality prediction, phenotyping, and length-of-stay prediction. Experiments are performed on both a base cohort and a matched subset, and the authors further analyze the effects of CXR modality missingness and subgroup performance across demographic and clinical factors (age, sex, race, and admission type).

**Strengths:**

- The benchmark incorporates new and clinically relevant features compared to existing state-of-the-art benchmarks, including cardiac rhythm, urine output, oxygen delivery, and ventilator settings.
- The experimental setup includes extensive hyperparameter tuning, which enhances methodological rigor.
- The paper is well-written and clearly structured, facilitating comprehension.
- The authors plan to open source their code.

**Weaknesses:**

- Figures should be numbered in the order of appearance in the main text. Supplementary figures should be explicitly labeled and referenced as such to avoid confusion.
- There appears to be a discrepancy between the statements “we restricted the chest X-ray (CXR) cohort to scans acquired during the patient’s current ICU stay” and “availability of at least one chest radiograph within a window spanning 24 hours before to 48 hours after ICU admission.” Given that certain tasks (e.g., phenotyping) rely on the full patient stay, it would be beneficial to clarify why later CXRs were excluded.
- The distinction between the base cohort and matched subset should be more clearly defined. Initially, it seems that the base cohort includes ICU patients only, yet multimodal baselines appear in Table 3, which suggests otherwise.
- Confidence intervals should be included for key performance metrics to provide a measure of statistical reliability.
- The manuscript would benefit from quantifying (1) the model-specific performance changes when transitioning from the paired to the partial dataset, and (2) the performance impact of incorporating the new EHR data features.
- The rationale behind using AUPRC thresholds of 0.5 and 0.45 for subgroup comparison in Insight 6 should be clarified. What is the significance or empirical justification for these specific values?
- Insight 7 is relatively weak. The results should be quantitatively supported, as the narrative observations are not fully consistent with the corresponding figure.

**Questions:**

Please see above.

While I appreciate the effort, I also have a question about novelty and contribution. This paper extends existing benchmarks in terms of analysis (missingness and fairness) and by adding a few more EHR features. Can the authors please explain why they think could be considered as significant contributions? The scope feels a bit limited for an audience at ICLR.

---

### Official Review · Reviewer_iaLb · 2025-11-10

**Soundness:** 3
**Presentation:** 3
**Contribution:** 2
**Rating:** 2
**Confidence:** 4

**Summary:**

The paper presents CareBench, a benchmark for multimodal fusion of EHR and chest X-rays built from MIMIC-IV and MIMIC-CXR. It standardizes a data pipeline, implements fifteen uni-modal and multi-modal models, and uses an evaluation protocol that covers accuracy, stability under missing modalities, and subgroup fairness. The dataset includes a base cohort where CXR is often missing, and a matched subset with paired EHR and CXR. The study evaluates phenotyping, in-hospital mortality, and remaining length of stay using a shared training setup and multiple metrics. Main findings are that fusion helps most when modalities are complete, that stability depends on the model family and the task, and that multimodality can widen fairness gaps for some attributes.

**Strengths:**

1. Cohort construction and preprocessing choices are documented with enough detail to be repeatable; the pipeline decisions (e.g., AP frontal views, avoiding treatment leakage, retaining missingness masks) are reasonable.

2. The robustness analysis in Figure 2 is informative: the shaded variance bands and per-task trends offer a clear picture of stability under increasing CXR missingness, supporting the claim that robustness is architecture- and task-dependent.

**Weaknesses:**

1. A central limitation is the lack of clinician adjudication for labels across all three tasks. Phenotypes come from ICD codes, mortality is based on discharge status, and remaining length of stay is a discretized target chosen by the authors. These sources are consistent and reproducible, yet they can include coding noise and do not confirm that targets match clinician judgment at the chart level. And do not guarantee that the benchmark reflects clinician-confirmed clinical states.


2. Recency is limited. The baseline pool spans well known fusion and missing-modality methods referenced through 2024. The baseline section and the configuration tables list methods such as UTDE, DAFT, MMTM, MedFuse, HEALNet, Flex-MoE, DrFuse, UMSE, ShaSpec, M3Care, and SMIL, with citations dated 2021 to 2024. No 2025 models appear in the comparisons in Tables 2 and 3, so the snapshot is not the newest possible view of the field at the time of writing.

3. Novelty in data is limited. The work assembles base and matched cohorts directly from MIMIC-IV and MIMIC-CXR rather than introducing new patient data or new labels, which narrows external validity to the same hospital system and collection procedures. Prior multimodal benchmarks had already considered missing data and fairness, for example MC-BEC and RadFusion, even if not in the same EHR–CXR setting.

**Questions:**

Please see the questions in the weaknesses above.

---

### Meta-Review · Area_Chair_CD1h · 2025-12-16

**Summary:**

No response was given to the weaknesses highlighted by the reviewers. I therefore consider that the authors agree with these criticisms and that they cannot be easily addressed.

**Reviewer Scores:**

N/A

---

### Decision · Program_Chairs · 2026-01-26

Reject